# NPAS4 in the medial prefrontal cortex mediates chronic social defeat stress-induced anhedonia-like behavior and reductions in excitatory synapses

Brandon W Hughes[1], Benjamin M Siemsen[1,2†], Evgeny Tsvetkov[1], Stefano Berto[1], Jaswinder Kumar[3,4], Rebecca G Cornbrooks[1], Rose Marie Akiki[1], Jennifer Y Cho[1], Jordan S Carter[1], Kirsten K Snyder[1], Ahlem Assali[1], Michael D Scofield[1,2], Christopher W Cowan[1,3,4]*, Makoto Taniguchi[1,3]*

[1]Department of Neuroscience, Medical University of South Carolina, Charleston, United States; [2]Department of Anesthesiology, Medical University of South Carolina, Charleston, United States; [3]Department of Psychiatry, Harvard Medical School, Belmont, United States; [4]Neuroscience Graduate Program, University of Texas Southwestern Medical Center, Dallas, United States

*For correspondence:
cowanc@musc.edu (CWC);
taniguch@musc.edu (MT)

Present address: †Supernus Pharmaceuticals, Inc, Rockville, United States

Competing interest: The authors declare that no competing interests exist.

**Abstract** Chronic stress can produce reward system deficits (i.e., anhedonia) and other common symptoms associated with depressive disorders, as well as neural circuit hypofunction in the medial prefrontal cortex (mPFC). However, the molecular mechanisms by which chronic stress promotes depressive-like behavior and hypofrontality remain unclear. We show here that the neuronal activity-regulated transcription factor, NPAS4, in the mPFC is regulated by chronic social defeat stress (CSDS), and it is required in this brain region for CSDS-induced changes in sucrose preference and natural reward motivation in the mice. Interestingly, NPAS4 is not required for CSDS-induced social avoidance or anxiety-like behavior. We also find that mPFC NPAS4 is required for CSDS-induced reductions in pyramidal neuron dendritic spine density, excitatory synaptic transmission, and presynaptic function, revealing a relationship between perturbation in excitatory synaptic transmission and the expression of anhedonia-like behavior in the mice. Finally, analysis of the mice mPFC tissues revealed that NPAS4 regulates the expression of numerous genes linked to glutamatergic synapses and ribosomal function, the expression of upregulated genes in CSDS-susceptible animals, and differentially expressed genes in postmortem human brains of patients with common neuropsychiatric disorders, including depression. Together, our findings position NPAS4 as a key mediator of chronic stress-induced hypofrontal states and anhedonia-like behavior.

## Editor's evaluation

This important manuscript shows compelling evidence for a role for the transcription factor NPAS4 in the medial prefrontal cortex in regulating stress-induced behavior, pyramidal neuron spine density, and gene expression. There is beautiful depth of mechanistic insight into how chronic stress produces anhedonia-like behavior. This paper will be of interest to the field of stress neurobiology and neuropsychiatry.

## Introduction

Stress-related mental disorders continue to be a leading cause of disability and financial burden on society (*Rehm and Shield, 2019*). The associated symptom domains of stress-related disorders are diverse and present with a high degree of comorbidity, thus treatment strategies for these disorders represent a major healthcare challenge. The rodent chronic social defeat stress (CSDS) paradigm produces multiple behavioral and neural phenotypes reminiscent of stress-related and depressive disorders in humans, including anhedonia-like behaviors and social avoidance (*Berton et al., 2006*; *Covington et al., 2010*; *Covington et al., 2009*; *Golden et al., 2011*; *Krishnan et al., 2007*; *Krishnan and Nestler, 2008*; *Venzala et al., 2012*; *Venzala et al., 2013*; *Vialou et al., 2014*; *Ye et al., 2016*). CSDS produces social avoidance in a subset of mice (i.e., stress-susceptible), whereas the resilient subpopulation displays normal social behavior and typically accounts for around 35–50% of the total population (*Krishnan et al., 2007*; *Krishnan and Nestler, 2008*; *Krishnan and Nestler, 2011*). Notably, these differences are analogous to human responses following chronic stress, where resilient individuals display greater optimism and cognitive flexibility, opposed to stress susceptibly increasing adverse responses to stress that can manifest as depression (*Dantzer et al., 2018*; *Han and Nestler, 2017*). Another CSDS-induced behavior is anhedonia, a core symptom of major depressive disorder (MDD) that is associated with deficits in hedonic capacity, reward evaluation, decision-making, and motivation to obtain rewards, as well as risk for suicide and treatment resistance (*Der-Avakian and Markou, 2012*; *Heshmati and Russo, 2015*; *Llorca and Gourion, 2015*; *Pizzagalli, 2014*; *Treadway and Zald, 2011*). Individuals who suffer from pathological stress often exhibit deficits in motivated, effort-based decision-making (*American Psychiatric Association, 2013*; *Chen et al., 2015*; *Henriques and Davidson, 2000*; *Pechtel et al., 2013*; *Porcelli and Delgado, 2017*), though clinical studies indicate that some individuals can exhibit positive behavioral outcomes following stress (i.e., stress resilience) (*Linley and Joseph, 2004*). Although these studies examined the differences in stress-related behaviors, including susceptibility vs. resilience, the neural mechanisms by which chronic stress produces anhedonia remain unclear. Multiple preclinical and clinical studies have revealed reduced function of the medial prefrontal cortex (mPFC), which is caused, at least in part, by stress-induced loss of structural and functional synaptic connections and circuits within this brain region (*Arnsten et al., 2015*; *Covington et al., 2005*; *Covington et al., 2010*; *Holmes and Wellman, 2009*; *Radley et al., 2006a*). Furthermore, chronic stress-induced hypofrontality is thought to underlie many symptoms of MDD (*Galynker et al., 1998*; *Llorca and Gourion, 2015*; *Matsuo et al., 2000*; *Suto et al., 2004*) and contribute to the neuropathology of treatment-resistant depression (*Li et al., 2015*), including the potential for anhedonia susceptibility (*Gong et al., 2018*; *Gong et al., 2017*).

In this study, we investigated the role of Neuronal PAS domain Protein 4 (NPAS4) in chronic stress-induced brain and behavior dysfunction. NPAS4 is an early response gene and transcription factor that modulates synaptic connections on excitatory (E) and inhibitory (I) neurons in response to synaptic activity – a proposed homeostatic mechanism to modulate E/I balance in strongly activated neural circuits (*Bloodgood et al., 2013*; *Brigidi et al., 2019*; *Lin et al., 2008*; *Sharma et al., 2019*; *Sim et al., 2013*; *Spiegel et al., 2014*; *Sun and Lin, 2016*). Previous studies have shown that *Npas4* KO mice have reduced anxiety (*Jaehne et al., 2015*), and that *Npas4* heterozygous mice have increased depression-like behavior in the forced swim test (*Shepard et al., 2019*). Stress exposure, including prenatal stress, maternal separation, restraint stress, and corticosterone administration in prenatal stages and adults, changes *Npas4* mRNA and protein expression in multiple brain regions (*Heslin and Coutellier, 2018*; *Yun et al., 2010*). However, the region-specific role of NPAS4 in the adult brain in response to stress is poorly understood. In the adult brain, NPAS4 is required in the hippocampus and amygdala for contextual fear learning (*Ploski et al., 2011*; *Ramamoorthi et al., 2011*), in the visual cortex for social recognition (*Heslin and Coutellier, 2018*), and in the nucleus accumbens (NAc) for cocaine reward-context learning and memory (*Taniguchi et al., 2017*). As such, NPAS4 is well-positioned to mediate adaptive cellular and synaptic changes produced by strong circuit activity, such as that produced in the mPFC by acute and chronic stress. Here, we discovered that acute and chronic social defeat stress induce NPAS4 expression in the mPFC, and that NPAS4 in this brain region is required for CSDS-induced anhedonia and attenuated excitatory input to mPFC pyramidal neurons, as well as reduced pyramidal neuron dendritic spine density. Similarly, we found that reducing mPFC *Npas4* alters expression of numerous downstream genes reported to be upregulated in stress-susceptible animals (*Bagot et al., 2016*) that are important for ribosome function

or excitatory synapse organization, activity, and signaling – the majority of which are differentially expressed in human patients with MDD (*Labonté et al., 2017*). Our findings revealed an essential role for a NPAS4 in chronic stress-induced mPFC hypofrontality and anhedonia-like behavior.

## Results

### Social defeat stress induces NPAS4 expression in the medial prefrontal cortex

We first characterized the cell type-specific *Npas4* mRNA expression in the mPFC, a key region associated with stress and reward, using a single-nuclei RNA-sequencing (snRNA-seq) approach. Consistent with the previous reports (*Lin et al., 2008*; *Spiegel et al., 2014*), *Npas4* is expressed only in the neurons, and we did not detect it in astrocytes or glial cells. *Npas4* mRNA is predominantly expressed in excitatory neurons (92.6%) throughout cortical layers 2 and 5/6, while a small fraction (7.4%) were found in multiple classes of GABAergic inhibitory neurons (7.4%), including *Adarrb2*-, *Pvalb*-, and *Sst*-positive neurons (*Figure 1A–D*). Also, 7% of all mPFC excitatory neurons expressed detectable *Npas4* mRNA, opposed to expression in only 2.5% of inhibitory neurons (*Figure 1D*). Next, we examined the expression of *Npas4* mRNA in two key corticolimbic regions, the mPFC and the nucleus accumbens (NAc), following 11 days of CSDS. We compared CSDS responses to a single social defeat stress experience (acute stress; *Figure 1E*). We observed a very rapid and transient induction of *Npas4* mRNA in the mPFC (*Figure 1F*, two-way ANOVA, F value = 16.6 and df = 77, Tukey's post hoc analysis: control vs. acute stress at 5 min, $p<0.0001$, control vs. chronic stress at 5 min, $p<0.0001$, acute vs. chronic stress at 5 min, $p<0.0001$, n = 9–10 per group, control vs. acute stress at 15 min, $p<0.0001$, control vs. chronic stress at 15 min, $p<0.0001$, acute vs. chronic stress at 15 min, $p<0.0001$, n = 6–10 per group) and NAc (*Figure 1—figure supplement 1A*). We observed a similar response with *cFos* mRNA in the mPFC, albeit a slower induction and longer duration of expression (*Figure 1—figure supplement 1B*). Interestingly, CSDS-induced expression of both *Npas4* and *cFos* was observed, although it was reduced compared to the acute stress response (*Figure 1F*, *Figure 1—figure supplement 1B*), possibly due to CSDS-induced mPFC hypofunction. In contrast, the CSDS-induced attenuation of *Npas4* induction was not observed in the NAc (*Figure 1—figure supplement 1A*). Analogous to stress-induced increases in *Npas4* mRNA, we observed a significant increase in NPAS4 protein at 1 hr following CSDS or acute stress exposure in multiple mPFC regions, including the anterior cingulate and prelimbic cortex subregions (*Figure 1G*, two-way ANOVA, F value = 9.695 and Df = 27, Tukey's post hoc analysis: control vs. acute stress in anterior cingulate cortex, $p=0.0267$, control vs. chronic stress in anterior cingulate cortex, $p=0.0462$, n = 3–5 per group, control vs. acute stress in prelimbic cortex, $p=0.0281$, control vs. chronic stress in prelimbic cortex, $p=0.0474$, n = 3–5 per group). Consistent with the snRNA-seq data (*Figure 1A–D*), the vast majority (>75%) of NPAS4+ neurons in the mPFC were co-expressed with CaMKIIα, a classical protein-marker for excitatory pyramidal neurons (*Figure 1H*, two-way ANOVA, F value = 5.645 and Df = 24, Tukey's post hoc analysis: control vs. acute stress in CaMKIIα(+) cells, $p=0.0131$, control vs. chronic stress in CaMKIIα(+) cells, $p<0.0001$, n = 8–11 per group), with practically no detectable NPAS4 expression in parvalbumin- or somatostatin-expressing GABAergic interneurons (*Figure 1I and J*). Similar to *Npas4* mRNA, the relative NPAS4 protein expression per cell was highest following acute stress (*Figure 1—figure supplement 1C*), suggesting that acute stress and CSDS activate a similar number of NPAS4-positive mPFC neurons, but the NPAS4 expression level within each cell is lower following repeated psychosocial stress.

### NPAS4 in the mPFC is required for CSDS-induced anhedonia-like behavior

To examine the function of NPAS4 in CSDS-induced behaviors (*Figure 2A*), we employed a neurotropic AAV-mediated RNA-interference approach to reduce endogenous *Npas4* in the mPFC using a prevalidated *Npas4* short hairpin RNA (shRNA) (AAV2-*Npas4* shRNA[PFC] and *Figure 2B*, paired *t*-test, t value = 3.7 and Df = 3, $p=0.0343$, n = 4 per group), which reliably reduces NPAS4 expression in multiple studies, and where knockdown effects have been repeatedly validated using *Npas4* conditional KO mice (*Lin et al., 2008*; *Maya-Vetencourt et al., 2012*; *Taniguchi et al., 2017*). Adult male mice (C57BL/6J) received a bilateral injection of AAV2-*Npas4* shRNA[PFC] or AAV2-shRNA scrambled control (AAV2-SC shRNA[PFC]). Mice were subjected to 10 days of CSDS or no stress control condition,

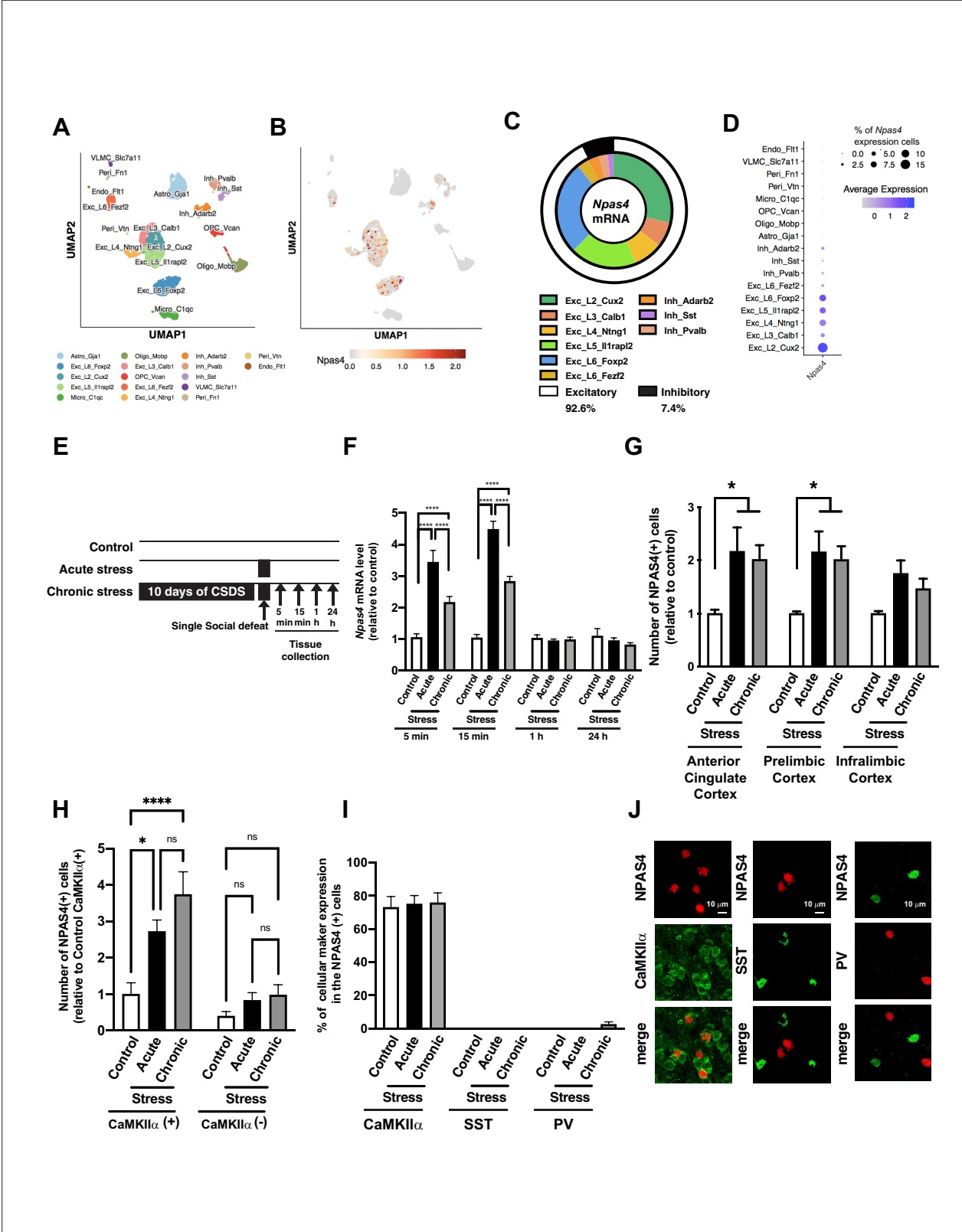

**Figure 1.** Social defeat stress induces NPAS4 expression in the medial prefrontal cortex (mPFC). (**A, B**) Uniform manifold approximation and projection (UMAP) plot of the mPFC single cells colored by cell type (**A**) and *Npsa4* mRNA expression (**B**). Cell types were defined by known markers and confirmed by predictive modeling using a single-cell mPFC atlas. (**C**) Donut chart represents the percentage of cell types that express *Npas4* mRNA. (**D**) Dot plot represents the percentage of *Npas4* mRNA expressing neurons in each cell type. (**E**) Schematic illustration of experimental timeline of gene

*Figure 1 continued on next page*

*Figure 1 continued*

expression analyses following acute social defeat stress and 10 days of chronic social defeat stress (CSDS). (**F**) Data plot represents the quantification of *Npas4* mRNA expression following acute and chronic social defeat stress at 5 min, 15 min, 1 hr, and 24 hr (n = 5–10/condition). (**G**) Quantification of fold change in NPAS4-positive cell number following acute and chronic social defeat stress in subregions of the mPFC, including the anterior cingulate, prelimbic, and infralimbic cortices (n = 3–5/condition). (**H**) Quantification of mPFC NPAS4-positive cells relative to the number of CaMKIIα-positive cells in control/no-stress mice. (**I, J**) Data plot shows the percentage of CaMKIIα-, somatostatin (SST)-, and parvalbumin (PV)-positive cells in NPAS4-positive cells within the mPFC after acute stress and CSDS (n = 3–9/condition), as well as representative IHC images of NPAS4 colocalization in these respective cell type. Scale bar, 10 µm. Data shown are mean ± SEM; *p<0.05, ****p<0.0001. Also see *Source data 1* for detailed statistical analyses.

The online version of this article includes the following source data and figure supplement(s) for figure 1:

**Source data 1.** *Figure 1F*.

**Source data 2.** *Figure 1G*.

**Source data 3.** *Figure 1H*.

**Source data 4.** *Figure 1I*.

**Figure supplement 1.** Social defeat stress induces NPAS4 and cFos expression in the nucleus accumbens (NAc) and medial prefrontal cortex (mPFC).

**Figure supplement 1—source data 1.** *Figure 1—figure supplement 1A*.

**Figure supplement 1—source data 2.** *Figure 1—figure supplement 1B*.

**Figure supplement 1—source data 3.** *Figure 1—figure supplement 1C*.

and then they were tested for sociability, natural reward preference and motivation, and anxiety-like behavior (*Figure 2A*). The CSDS-treated SC shRNA[PFC] and *Npas4* shRNA[PFC] mice showed a significant reduction in the time spent interacting with a novel social target, as shown by time spent in the interaction zone in the presence a social target (*Figure 2C*, SC shRNA[PFC] mice, two-way ANOVA, F value = 6.69 and Df = 41, Bonferroni post hoc analysis, interaction partner (-) vs. (+) in control no stress animals, p<0.0001, interaction partner (-) vs. (+) in CSDS animals, p=0.0099, control no stress animals vs. CSDS animals in interaction partner (+), p=0.0005, n = 18–25 per group, *Npas4* shRNA[PFC] mice, two-way ANOVA, F value = 7.553 and Df = 39, Bonferroni post hoc analysis, interaction partner (-) vs. (+) in control no stress animals, p<0.0001, interaction partner (-) vs. (+) in CSDS animals, p=0.0021, control no stress animals vs. CSDS animals in interaction partner (+), p=0.0034, n = 19–22 per group). In addition, there was a main effect of CSDS, but no significant difference between *Npas4* shRNA[PFC] vs. SC shRNA[PFC] mice in the relative distribution of social interaction ratio in CSDS-treated mice (*Figure 2D*, two-way ANOVA, main effect of CSDS, F value = 10.01 and Df = 78, p=0.0022, n = 18–25). Both *Npas4* shRNA[PFC] and SC shRNA[PFC] mice showed significantly increased social avoidance time and ratio following CSDS (*Figure 2E and F*; *Figure 2E*, SC shRNA[PFC] mice, two-way ANOVA, F value = 5.541 and Df = 38, Bonferroni post hoc analysis, control no stress animals vs. CSDS animals in interaction partner (+), p<0.0001, n = 16–24 per group, *Npas4* shRNA[PFC] mice, two-way ANOVA, F value = 4.666 and Df = 38, Bonferroni post hoc analysis, control no stress animals vs. CSDS animals in interaction partner (+), p=0.0015, n = 19–21 per group; *Figure 2F*, two-way ANOVA, main effect of CSDS, F value = 15.64 and Df = 76, p=0.0002, n = 16–24 per group), suggesting that mPFC NPAS4 is not required for CSDS-induced social avoidance. However, unlike the CSDS-treated SC shRNA[PFC] mice, CSDS-treated *Npas4* shRNA[PFC] mice did not develop anhedonia-like behavior, as detected by a significant reduction in sucrose preference in the two-bottle choice test (*Figure 2G* and *Figure 2—figure supplement 1A*; *Figure 2G*, two-way ANOVA, F value = 5.548 and Df = 65, Tukey's post hoc analysis, control no stress vs. CSDS in SC shRNA[PFC] mice, p=0.0291, SC shRNA[PFC] vs. *Npas4* shRNA[PFC] mice with CSDS, p=0.0492, n = 11–24 per group). Interestingly, CSDS increased anxiety-like behavior, as measured in the elevated plus maze, in both SC shRNA[PFC] and *Npas4* shRNA[PFC] mice (*Figure 2H*, two-way ANOVA, main effect of CSDS, F value = 8.087 and Df = 59, p=0.0061, n = 14–18 per group), indicating that mPFC NPAS4 function is required for some, but not all, of the behavioral sequelae of CSDS. These data suggest that the molecular and circuit mechanisms of CSDS-induced social avoidance, anhedonia, and anxiety might be distinct. Moreover, the presence of CSDS-induced social avoidance and anxiety-related behavior in *Npas4* shRNA[PFC] mice argues against the possibility that they are simply less sensitive to stress and/or have deficits in threat/fear-related learning and memory.

Individuals who suffer from pathological stress often exhibit reduced motivation to pursue rewards (*American Psychiatric Association, 2013*; *Chen et al., 2015*); however, it is also commonly reported that a subset of individuals can exhibit positive behavioral outcomes following stress (i.e., stress

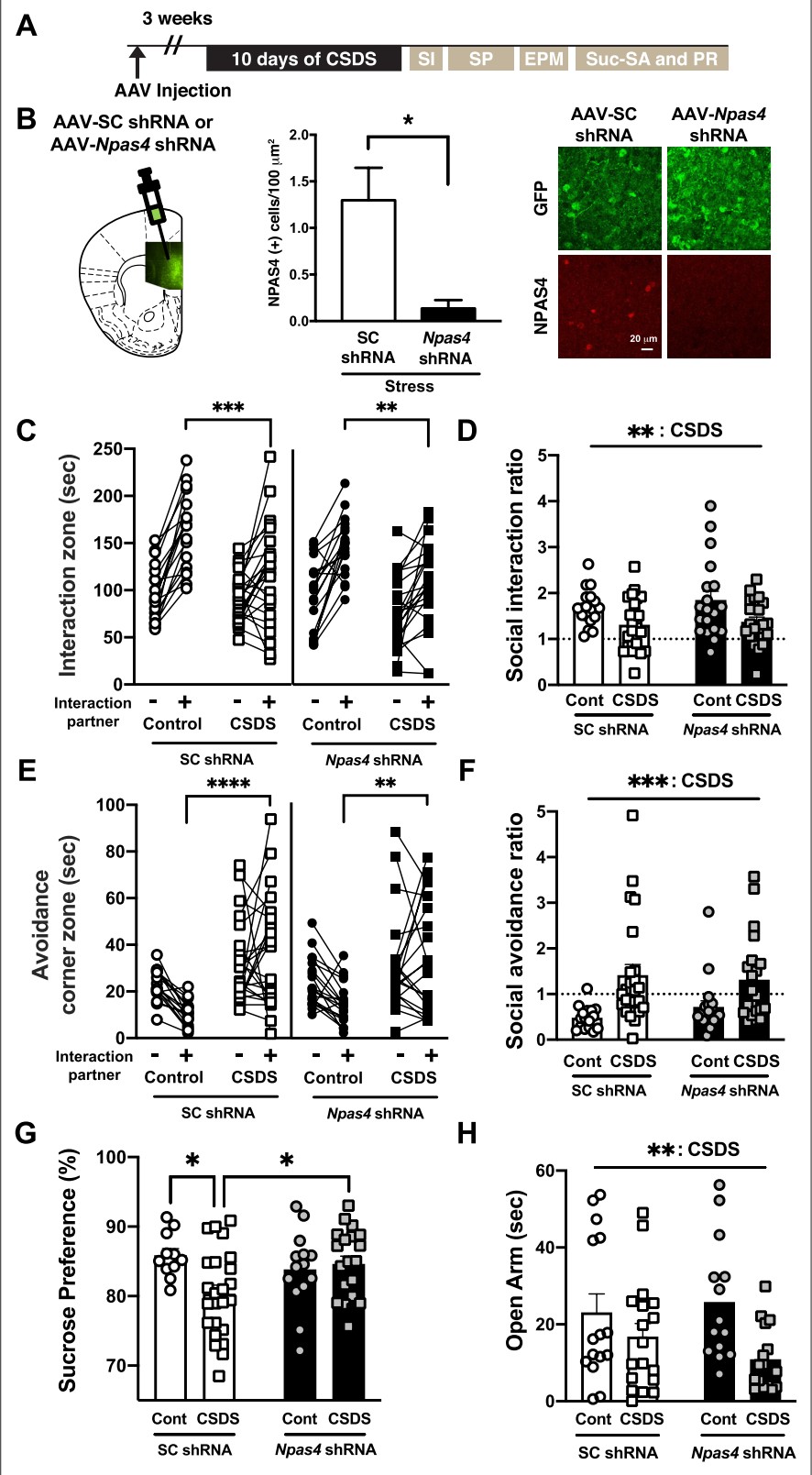

**Figure 2.** NPAS4 in the medial prefrontal cortex (mPFC) is required for chronic social defeat stress (CSDS)-induced anhedonia-like behavior. (**A**) Schematic illustration of experimental timeline of behavioral test battery consisting of CSDS followed by social interaction (SI; **C–F**), sucrose preference (SP; **G**), elevated plus maze (EPM; **H**), sucrose self-administration, and progressive ratio testing (Suc-SA and PR; **Figure 3A–D**). (**B**) AAV2-*Npas4* shRNA in the

*Figure 2 continued on next page*

*Figure 2 continued*

adult male mPFC decreases stress-induced NPAS4 protein expression. Left: representative image showing AAV2-shRNA expression viral vector-mediated eGFP expression in the adult mice mPFC. Right: quantification of NPAS4-positive cells/100 µm² (n = 4/condition). (**C**) and (**D**) CSDS decreases the time spent in the social interaction zone (**C**) and the social interaction ratio (**D**) in SC shRNA$^{PFC}$ and *Npas4* shRNA$^{PFC}$ mice after CSDS (n = 18–25/condition). (**E**) and (**F**) CSDS increases the time spent in the avoidance corner zone and social avoidance ratio in SC shRNA$^{PFC}$ and Npas4 shRNA$^{PFC}$ mice (n = 16–24/condition). (**G**) CSDS-induced reduction of sucrose preference is blocked by *Npas4* shRNA in the mPFC (**F**; n = 11–24). (**H**) CSDS reduces time spent in open arms (sec) in SC shRNA$^{PFC}$ and *Npas4* shRNA$^{PFC}$ mice (n =14–18).

The online version of this article includes the following source data and figure supplement(s) for figure 2:

**Source data 1.** *Figure 2B*.

**Source data 2.** *Figure 2C*.

**Source data 3.** *Figure 2D*.

**Source data 4.** *Figure 2E*.

**Source data 5.** *Figure 2F*.

**Source data 6.** *Figure 2G*.

**Source data 7.** *Figure 2H*.

**Figure supplement 1.** NPAS4 in the medial prefrontal cortex (mPFC) is required for chronic social defeat stress (CSDS)-induced reduction of sucrose consumption.

**Figure supplement 1—source data 1.** *Figure 2—figure supplement 1*.

resilience) (***Linley and Joseph, 2004***). To examine the role of NPAS4 in CSDS-induced changes in reward motivation, *Npas4* shRNA$^{PFC}$ or SC shRNA$^{PFC}$ mice were subjected to CSDS or the 'no stress' condition, and then they were allowed to self-administer sucrose (sucrose SA) under operant conditions. After stable sucrose SA was established, we examined motivation to work for a sucrose reward using the progressive ratio (PR) schedule of reinforcement. Compared to SC shRNA$^{PFC}$ controls, *Npas4* shRNA$^{PFC}$ mice displayed no differences in acquisition of sucrose SA (*Figure 3A*) or operant discrimination learning (nosepokes in the active vs. inactive port) (*Figure 3B*). Interestingly, *Npas4* shRNA$^{PFC}$ significantly increased PR breakpoint – the maximum number of nose-pokes an animal was willing to perform to receive a single sucrose reward (*Figure 3C*, two-way ANOVA, main effect of shRNA expression, F value = 5.92 and Df = 58, p=0.0181, n = 13–19 per group), suggesting that reducing levels of mPFC NPAS4 might enhance reward motivation. Notably, in animals susceptible to CSDS, *Npas4* shRNA significantly increased motivation to obtain sucrose, with no change in PR breakpoint after *Npas4* shRNA in resilient animals (*Figure 3D*, two-way ANOVA, F value = 5.685 and Df = 31, Bonferroni post hoc analysis, SC shRNA$^{PFC}$ and *Npas4* shRNA$^{PFC}$ mice in susceptible group, p=0.0353, n = 3–14 per group), suggesting that CSDS-induced mPFC NPAS4 influences natural reward motivation.

## NPAS4 regulates CSDS-induced reductions in mPFC dendritic spine density and excitatory synaptic transmission

CSDS-induced reduction of dendritic spine density on mPFC pyramidal neurons is a putative pathophysiological underpinning of depression-associated behavior (***Cerqueira et al., 2007***; ***Colyn et al., 2019***; ***Liston et al., 2006***; ***McKlveen et al., 2013***; ***Ota and Duman, 2013***; ***Qiao et al., 2016***; ***Qu et al., 2018***; ***Shu and Xu, 2017***). As such, we quantified dendritic spine density on deep-layer pyramidal neurons in SC shRNA$^{PFC}$ or *Npas4* shRNA$^{PFC}$ mice after CSDS compared to nonstressed mice. As expected, we observed a CSDS-induced reduction in dendritic spine density in SC shRNA control mice (*Figure 4A*, top; *Figure 4B*, left). In contrast, we observed no CSDS-induced changes in mPFC dendritic spine density in *Npas4* shRNA$^{PFC}$ mice (*Figure 4A*, bottom; *Figure 4B*, right), suggesting that NPAS4, either directly or indirectly, is required for this chronic stress-induced structural synaptic change in the mPFC (*Figure 4B*, two-way ANOVA, F value = 9.864 and Df = 162, Tukey's post hoc analysis, control no stress vs. CSDS in SC shRNA$^{PFC}$ mice, p=0.0056, SC shRNA$^{PFC}$ vs. *Npas4* shRNA$^{PFC}$ mice after CSDS, p<0.0001, n = 34–55 dendrites/8 animals per group). Of note, no changes in mPFC dendritic spine density were observed in nonstressed *Npas4* shRNA$^{PFC}$ mice (*Figure 4A and B*), indicating that steady-state dendritic spine density in adult mPFC pyramidal neurons of nonstressed animals does not require normal NPAS4 expression levels. In addition, neither *Npas4* shRNA nor CSDS

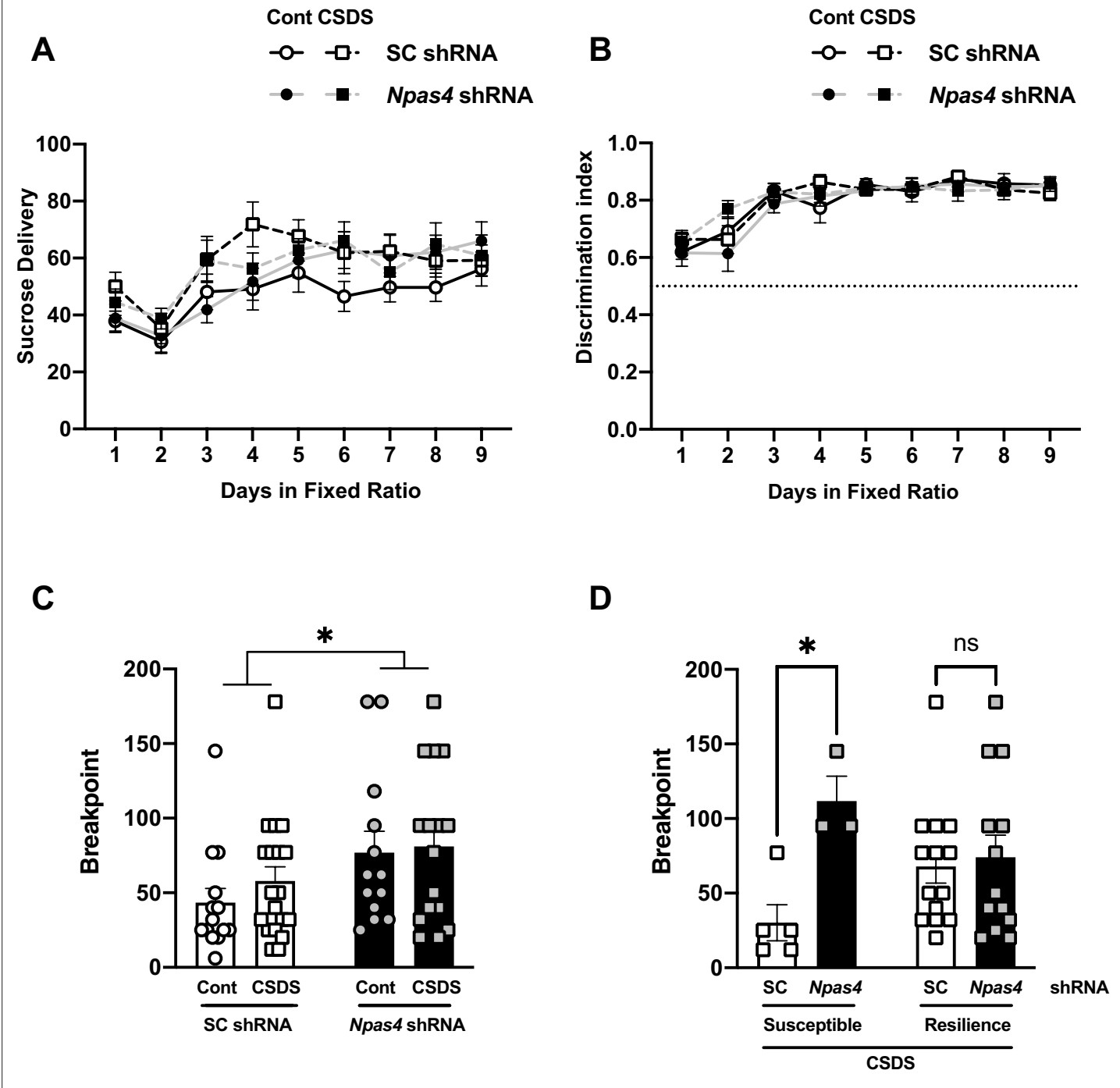

**Figure 3.** NPAS4 in the medial prefrontal cortex (mPFC) regulates effort-based motivated behavior during sucrose SA following chronic social defeat stress (CSDS). (**A, B**) Data plots showing the acquisition period of sucrose self-administration in SC shRNA[PFC] and *Npas4* shRNA[PFC] mice after CSDS or no stress control condition, with no change in the number of sucrose delivery (**A**) and in the discrimination ratio between the active and inactive nosepokes (**B**; n = 14–18/group). (**C**) Data plot showing the maximum number of active nose pokes required to receive a sucrose reward (breakpoint) after CSDS in the PR test of both SC shRNA[PFC] and *Npas4* shRNA[PFC] mice. *Npas4* shRNA[PFC] mice demonstrated a significantly higher PR breakpoint compared to control SC shRNA[PFC] mice (n = 13–19/group). (**D**) *Npas4* shRNA[PFC] mice susceptible, but not resilience, to CSDS demonstrated a significantly higher breakpoint compared to SC shRNA[PFC] mice after CSDS (n =3–14/group).

The online version of this article includes the following source data for figure 3:

**Source data 1.** *Figure 3A*.

**Source data 2.** *Figure 3B*.

*Figure 3 continued on next page*

*Figure 3 continued*

**Source data 3.** *Figure 3C*.

**Source data 4.** *Figure 3D*.

produced any detectable changes in mean dendritic spine head diameter or distribution (*Figure 4—figure supplement 1*).

Rodent models show that chronic restraint, unpredictable (*Yuen et al., 2012*), or social defeat stress (*Kuang et al., 2022*) decreases excitatory transmission onto mPFC pyramidal neurons. Furthermore, the administration of ketamine increases excitatory transmission in cultured neurons in vitro (*Gerhard et al., 2020*) and in mPFC pyramidal neurons in vivo (*Zhang et al., 2020*), and it alleviates symptoms of depression in human MDD patients through increased mPFC activity (*Hare and Duman, 2020*). In line with these data and our findings on CSDS-induced decreases in dendritic spine density (*Figure 4B*), CSDS in control animals significantly increased the miniature excitatory postsynaptic current (mEPSC) inter-event interval in the layer 5 mPFC pyramidal neurons of SC shRNA$^{PFC}$ mice (*Figure 4C and D*), consistent with a decrease in presynaptic function and/or reduction in synapse number. However, this change in mEPSC frequency was absent in the CSDS-treated *Npas4* shRNA$^{PFC}$ mice (*Figure 4C and D*; *Figure 4C*, two-way ANOVA, F value = 14.57, and Df = 4251, Tukey's post hoc analysis, control no stress vs. CSDS in SC shRNA$^{PFC}$ mice, p<0.0001, SC shRNA$^{PFC}$ vs. *Npas4* shRNA$^{PFC}$ mice after CSDS, p<0.0001, n = 648–1240 events/6–12 neurons/2–4 animals per group). Notably, *Npas4* shRNA$^{PFC}$ also produced a significant increase in mEPSC amplitude in mPFC pyramidal neurons (*Figure 4E–G*; main effect of *Npas4* shRNA), but CSDS did not have this effect (*Figure 4E–G*; *Figure 4E*, two-way ANOVA, F value = 6.992 and Df = 4301, Tukey's post hoc analysis, SC shRNA$^{PFC}$ vs. *Npas4* shRNA$^{PFC}$ mice with control no stress, p<0.0001, SC shRNA$^{PFC}$ vs. *Npas4* shRNA$^{PFC}$ mice with CSDS, p<0.0001, n = 654–1253 events/6–12 neurons/2–4 animals per group), suggesting that NPAS4 limits glutamatergic synaptic strength on mPFC pyramidal neurons. Finally, CSDS in control animals significantly increased the paired-pulse ratio (PPR) in excitatory pyramidal neurons, suggesting a reduction presynaptic release probability, but this CSDS-induced effect on presynaptic function was blocked by *Npas4* shRNA (*Figure 4H*, Two-way ANOVA, F value = 5.883 and Df = 561, Tukey's post hoc analysis, control no stress vs. CSDS in SC shRNA$^{PFC}$ mice, p=0.0002, SC shRNA$^{PFC}$ vs. *Npas4* shRNA$^{PFC}$ mice with CSDS, p<0.0056, n = 133–191 events/10–17 neurons/3–5 animals per group). Together, our data reveal that NPAS4 in mPFC is required for reductions in excitatory synaptic transmission and synapse density following chronic psychosocial stress, and that NPAS4 limits basal glutamatergic synaptic strength of deep-layer pyramidal neurons.

## NPAS4 regulates the expression of ribosomal and glutamatergic synapse genes

To analyze the influence of NPAS4 on the mPFC transcriptome, we performed RNA-seq analyses with mPFC tissue isolated from SC shRNA$^{PFC}$ and *Npas4* shRNA$^{PFC}$ mice. Of the ~700 differentially expressed genes (DEGs, FDR < 0.05, log$_2$ (FC) > |0.3|) following *Npas4* mRNA knockdown in mPFC, 267 were downregulated and 365 were upregulated (*Supplementary file 1*). Downregulated genes included *Spata3*, *Defb1*, *Cidea*, *Psmb10,* and *Rspo3* and upregulated genes included *Arc*, *Igfn1*, *Schip1*, *Apcdd1*, and *Dapk2* (*Figure 5A and B*). A subset of these DEGs was independently validated by qRT-PCR using independent mPFC samples isolated from SC shRNA$^{PFC}$ and *Npas4* shRNA$^{PFC}$ mice 1 hr after acute social defeat or control, no stress conditions, including *Npas4*;Two-way ANOVA, F value = 6.736 and Df = 24, Tukey's post hoc analysis, control SC shRNA vs. *Npas4* shRNA mice after acute social defeat, p<0.0159, n = 7 animals per group, *Ache* (acetylcholinesterase; two-way ANOVA, main effect of *Npas4* shRNA, F value = 20 and Df = 24, p=0.0002, n = 7 animals per group), *Arpp21* (cAMP regulated phosphoprotein 21; two-way ANOVA, main effect of *Npas4* shRNA, F value = 7.433 and Df = 24, p=0.0118, n = 7 animals per group), *Dhcr7* (7-dehydrocholesterole reductase; two-way ANOVA, main effect of *Npas4* shRNA, F value = 10 and Df = 24, p=0.0042, n = 7 animals per group), *Hps4* (HPS4 biogenesis of lysosomal organelles complex 3 subunit 2; two-way ANOVA, main effect of *Npas4* shRNA, F value = 7.36 and Df = 24, p=0.0121, n = 7 animals per group), *Nfix* (nuclear factor I X; two-way ANOVA, main effect of *Npas4* shRNA, F value = 6.568 and Df = 24, p=0.0171, n = 7 animals per group), and *Sst* (somatostatin; two-way ANOVA, main effect of *Npas4* shRNA, F

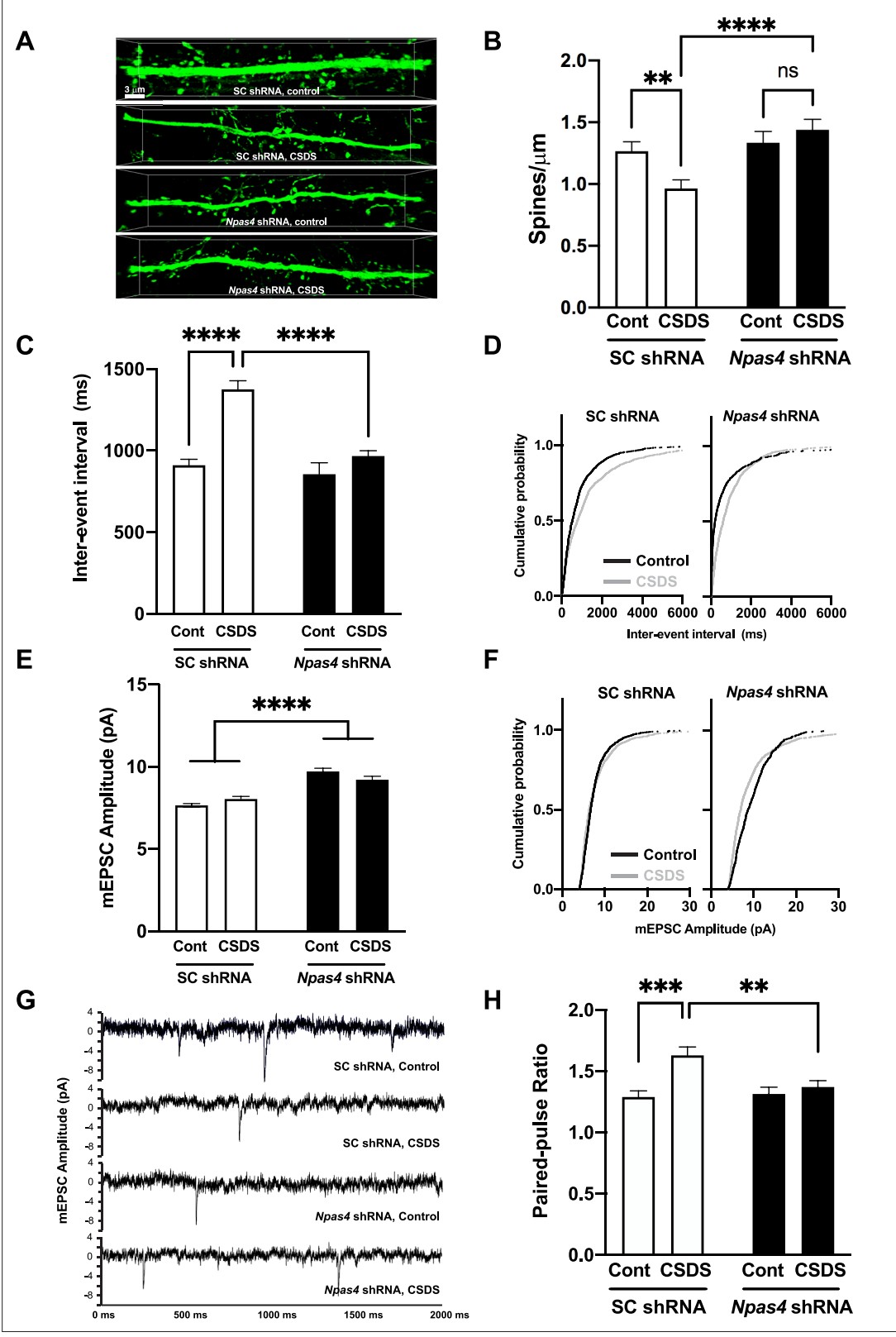

**Figure 4.** NPAS4 regulates chronic social defeat stress (CSDS)-induced reductions in medial prefrontal cortex (mPFC) dendritic spine density and excitatory synaptic transmission. (**A, B**) NPAS4 regulates CSDS-induced reduction of dendritic spine density in the mPFC. (**A**) Representative images showing AAV2-shRNA expression viral vector-mediated eGFP expression. Scale bar, 3 μm. (**B**) Quantification of dendritic spine density of deep layer mPFC pyramidal neurons from SC shRNAPFC and *Npas4* shRNA[PFC] mice after CSDS or in no stress controls (n = 34–55 branch/8 animals/condition).

*Figure 4 continued on next page*

*Figure 4 continued*

(**C**) Inter-event interval after *Npas4* knockdown and CSDS. (**D**) Cumulative probability of inter-event interval after CSDS after SC shRNA[PFC] and *Npas4* shRNA[PFC]. (**E**) Miniature excitatory postsynaptic current (mEPSC) amplitude after *Npas4* knockdown and CSDS. (**F**) Cumulative probability of mEPSCC amplitude after CSDS after SC shRNA[PFC] and *Npas4* shRNA[PFC]. (**G**) Representative mEPSC traces. (**H**) Paired-pulse ratio recordings after *Npas4* knockdown and CSDS. Data shown are mean ± SEM; *p<0.05, ***p<0.001. Also see **Source data 1** for detailed statistical analyses.

The online version of this article includes the following source data and figure supplement(s) for figure 4:

**Source data 1.** *Figure 4B*.

**Source data 2.** *Figure 4C*.

**Source data 3.** *Figure 4D*.

**Source data 4.** *Figure 4E*.

**Source data 5.** *Figure 4E*.

**Source data 6.** *Figure 4H*.

**Figure supplement 1.** Medial prefrontal cortex (mPFC) dendritic spine morphological analyses in the mPFC of SC shRNA[PFC] and *Npas4* shRNA[PFC] mice after chronic social defeat stress (CSDS).

**Figure supplement 1—source data 1.** *Figure 4—figure supplement 1*.

value = 5.496 and Df = 23, p=0.0281, n = 6–7 animals per group) (*Figure 5—figure supplement 1*). Interestingly, *Npas4* shRNA upregulated DEGs were significantly enriched in the Midnightblue (MB) module of DEGs that was identified by Bagot and colleagues (*Figure 5C*, top) (*Bagot et al., 2016*). This module consists of genes that are upregulated in the PFC of resilient mice, and gene ontology (GO) analysis showed significant enrichment of cell–cell signaling and synaptic transmission genes (*Bagot et al., 2016*). Furthermore, *Npas4* shRNA DEGs were significantly enriched in two PsychEN-CODE modules (*Figure 5C*, bottom) (*Gandal et al., 2018*; *Wang et al., 2018*); *Npas4* shRNA-downregulated DEGs showed significant enrichment within gene module M15, an excitatory neuron module of genes that are associated with ribosome function and upregulated in Autism Spectrum Disorder (ASD) and Bipolar Disorder (BD), while *Npas4* shRNA-upregulated DEGs showed significant enrichment in gene module M1, an excitatory neuron module of downregulated genes in ASD that are linked to glutamate-driven neuronal excitability (*Gandal et al., 2018*). Additionally, functional pathway analysis of *Npas4* shRNA-downregulated DEGs revealed significant enrichment of genes linked to ribosome function and protein synthesis. *Npas4* shRNA-upregulated DEGs showed significant enrichment of glutamatergic synapse-related genes important for synaptic signaling and organization (*Figure 5D*). To determine whether these mPFC DEGs are putative direct targets of NPAS4, we compared our data to previously published NPAS4 ChIP-seq studies (*Brigidi et al., 2019*; *Kim et al., 2010*) and found significant genomic enrichment of NPAS4 binding to promoter, intron, exon, and distal intergenic genomic regions (*Figure 5E*), suggesting that NPAS4 may directly regulate several key mPFC genes involved in the regulation of glutamatergic synapses and ribosomes. Interestingly, RNA-seq from human postmortem brains (BA8/9) of male MDD patients indicated significant enrichment of differentially expressed genes (p<0.05) in ribosome-related pathways, including 57 significantly upregulated genes (*Figure 5F*; *Labonté et al., 2017*). Of note, the enrichment of ribosomal genes was not observed in female MDD brains (BA8/9), where only one gene, *RPS28*, exhibited significant differential expression. Moreover, the majority (66.2%) of the *Npas4* shRNA-downregulated genes from our analysis overlapped with upregulated genes in human MDD patients (*Figure 5F*), suggesting that *Npas4* expression could contribute to vulnerability to depression in the human brain. Finally, NPAS4 ChIP-seq in hippocampal neurons (*Brigidi et al., 2019*) indicates that NPAS4 directly associates with 55% (42 of 77) of ribosome-related genes classified in the pathway 'co-translational protein targeting membrane' (*Figure 5G*). We detected 92 total ribosome-related genes in the mPFC that are classified in this pathway, with 68 downregulated and 2 upregulated (p<0.05) by *Npas4* shRNA (*Figure 5G*). Finally, we used qRT-PCR to validate several mPFC genes regulated by *Npas4* shRNA and acute social defeat stress and found that *Npas4* itself was the only regulated transcript at 1 hr following acute social defeat stress (*Figure 5—figure supplement 1*). Together, our data suggest that mPFC NPAS4 regulates numerous genes related to glutamatergic synapse regulation and ribosomal function and positions it as a key regulator of healthy mPFC function.

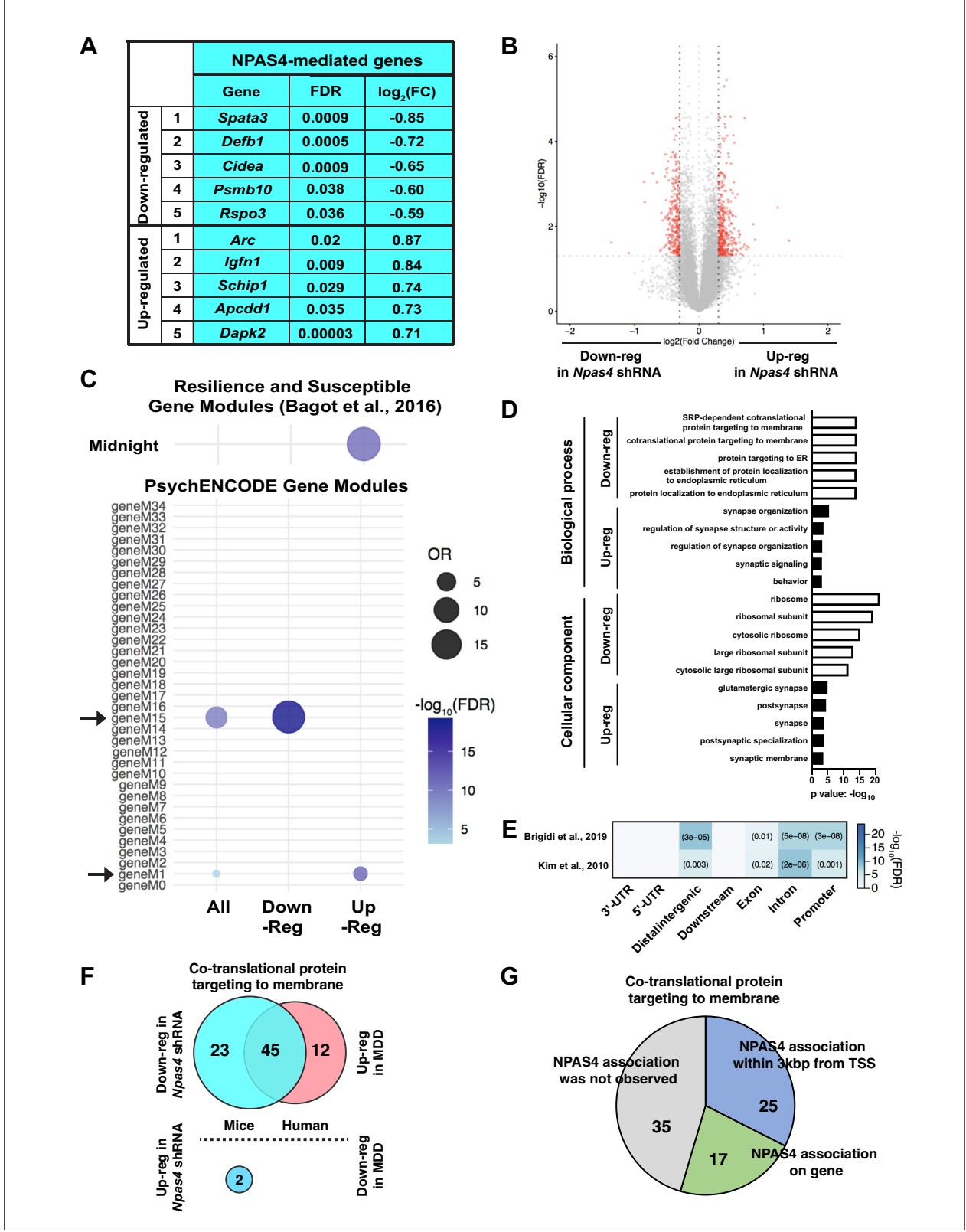

**Figure 5.** NPAS4 regulates the expression of ribosomal and glutamatergic synapse genes. (**A**, **B**) List of top differentially expressed genes in medial prefrontal cortex (mPFC) of *Npas4* shRNAPFC mice (**A**) and corresponding volcano plot of all significant DEGs (FDR < 0.05, log2 (FC) > |0.3|, red) compared to those that were not significant (gray; **B**). (**C**) *Npas4* DEG enrichment in gene modules that are deferentially regulated in Resilience and Susceptible animals in ***Bagot et al., 2016*** and are dysregulated in neuropsychiatric disorders; Modules M1 and M15, as shown by PsychENCODE.

*Figure 5 continued on next page*

*Figure 5 continued*

(**D**) Gene ontology analysis of down- and upregulated DEGs in *Npas4* shRNA[PFC] mice. (**E**) Comparison of mPFC genes regulated by *Npas4* shRNA[PFC] compared to previously published Npas4 ChIP-seq data (*Kim et al., 2010*; *Brigidi et al., 2019*). (**F**) Overlap of significantly differential expression genes (p<0.05) in *Npas4* shRNA[PFC] mice (left; blue) and differential expression genes (p<0.05) in BA8/9 of human major depressive disorder (MDD) patients (right; pink). (**G**) ChIP-seq analysis of NPAS4 association with significant ribosome-related differential expression genes identified from this study.

The online version of this article includes the following source data and figure supplement(s) for figure 5:

**Source data 1.** *Figure 5A*.

**Figure supplement 1.** Differential expression genes in the medial prefrontal cortex (mPFC) of *Npas4* shRNA mice.

**Figure supplement 1—source data 1.** *Figure 5—figure supplement 1*.

## Discussion

Here we find that social defeat stress (acute or chronic) induces rapid and transient expression of NPAS4 in mPFC neurons, and that NPAS4 in the mPFC is required for CSDS-induced anhedonia-like behavior, changes in effort-based reward seeking-motivated behavior, and CSDS-induced dendritic spine loss and suppression of excitatory synaptic transmission on mPFC pyramidal neurons. However, mPFC NPAS4 was not required for CSDS-induced social avoidance or anxiety-like behavior, suggesting that CSDS produces those phenotypes through distinct molecular and/or circuit mechanisms devoid of NPAS4 function. We found that NPAS4 influences the expression of hundreds of mPFC genes, including upregulated genes reported in stress-resilient animals and genes linked to glutamatergic synapses. As such, CSDS-induced NPAS4 could directly or indirectly downregulate these synapse-related genes and facilitate reductions in mPFC excitatory synaptic transmission. We also detected strong enrichment of downregulated ribosomal genes, many of which are also dysregulated in human MDD, suggesting that ribosomal gene dysregulation could be potential biomarkers of depression. Together, our findings reveal a novel and essential role for NPAS4 in chronic stress-induced anhedonia-like behavior and suppression of mPFC excitatory synaptic function.

NPAS4 is a neuronal-specific, synaptic activity-regulated transcription factor that regulates excitatory/inhibitory synapse balance and synaptic transmission (*Bloodgood et al., 2013*; *Brigidi et al., 2019*; *Hartzell et al., 2018*; *Lin et al., 2008*; *Spiegel et al., 2014*; *Sun and Lin, 2016*). Synaptic activity-dependent induction of NPAS4 in pyramidal neurons reduces excitatory synaptic transmission onto these neurons (*Lin et al., 2008*) and decreases excitatory synaptic inputs (*Sim et al., 2013*), consistent with our finding that NPAS4 is required for CSDS-induced loss of mPFC pyramidal neuron dendritic spine density and reduction of excitatory synaptic transmission. While one report indicated that CSDS-induced reduction of dendritic spine density is associated with social avoidance phenotypes (*Qu et al., 2018*), we observed that mPFC NPAS4 reduction selectively blocked CSDS-induced spine loss and anhedonia-like behavior, but social avoidance and anxiety-like behavior were not impacted, suggesting that deep-layer mPFC pyramidal cell spine loss, per se, is not strictly required for CSDS-induced social- and anxiety-related phenotypes.

Notably, we found that CSDS increased the mEPSC inter-event interval in mPFC pyramidal neurons in SC shRNA[PFC] control mice, which is consistent with reported effects of chronic restraint or unpredictable stress (*Yuen et al., 2012*). In contrast, Yuen et al. demonstrated that chronic stress also decreased mEPSC amplitude (*Yuen et al., 2012*), which we did not observe following CSDS, suggesting possible model-specific differences in mPFC neuroadaptations and highlighting the considerable heterogeneity of stress biology (*Duman et al., 2016*). In the future, it would be interesting to examine the role of NPAS4 in the other aversive experience-induced (e.g., chronic restraint or unpredictable stress) changes in mPFC pyramidal neuron excitatory synaptic transmission and depression-like behavior. In addition, mPFC NPAS4 mediates CSDS-induced reduction of glutamatergic presynaptic function (i.e., increased PPR), which could be a non-cell-autonomous effect of NPAS4 on long-range inputs to the mPFC deep-layer pyramidal neurons. It is interesting to note that the stress-independent increase in mEPSC amplitude produced by *Npas4* shRNA might produce a preexisting mPFC hyperfunction that protects the mPFC pyramidal neurons from CSDS-induced effects. Future studies will be important for understanding precisely how mPFC NPAS4 promotes mPFC hypofunction and anhedonia-like behavior, and whether therapeutic interventions, such as ketamine or antidepressant treatment, intersect with NPAS4-dependent mechanisms of stress-induced neuronal plasticity.

Although we targeted both the prelimbic and infralimbic subregions of the mPFC, studies show that these subregions can differentially regulate reward-related behavior (*Capuzzo and Floresco, 2020*; *Riaz et al., 2019*). As such, future studies examining the role of NPAS4 in these mPFC subregions following CSDS might provide valuable insights into NPAS4's influence on anhedonia-like behavior. Additionally, although we were unable to study females in our CSDS model, chronic exposure to stress hormones, chronic mild unpredictable stress, and chronic restraint stress all induce anhedonia-like behavior and dendritic spine loss in both sexes (*Brown et al., 2005*; *Christoffel et al., 2011*; *Cook and Wellman, 2004*; *Goldwater et al., 2009*; *Liston et al., 2006*; *Mayanagi and Sobue, 2019*; *Qiao et al., 2016*; *Radley et al., 2006a*; *Radley et al., 2005*; *Radley et al., 2006b*; *Radley et al., 2004*). Moreover, PFC pyramidal cell dendritic spine density is also reduced in human postmortem brains of individuals diagnosed with anhedonia-associated neuropsychiatric disorders, such as SCZ, BD, and MDD (*Christoffel et al., 2011*; *Duman and Duman, 2015*; *Forrest et al., 2018*; *Glausier and Lewis, 2013*; *Holmes et al., 2019*; *Konopaske et al., 2014*; *Lewis and González-Burgos, 2008*; *Moda-Sava et al., 2019*; *Moyer et al., 2015*; *Qiao et al., 2016*). These data support the functional relationship between excitatory neuronal transmission onto mPFC pyramidal neurons and anhedonia, and suggest the effects shown here are not sex-specific. However, future studies in female mice will be essential to interrogate this hypothesis.

NPAS4 in cultured neurons regulates a large, cell type-specific program of gene expression, including key targets like brain-derived neurotropic factor (BDNF), that alter E/I synapse balance (*Bloodgood et al., 2013*; *Lin et al., 2008*; *Ramamoorthi et al., 2011*; *Sim et al., 2013*; *Spiegel et al., 2014*; *Sun and Lin, 2016*; *Ye et al., 2016*). However, social defeat stress failed to induce *Bdnf* mRNA in mPFC and *Npas4* knockdown did not alter basal mPFC *Bdnf* expression (data not shown and *Supplementary file 1*), suggesting that *Bdnf* is not a key downstream target of mPFC NPAS4 in the context of CSDS. Our RNA-seq analysis of mPFC tissues, with or without *Npas4* shRNA, revealed an abundance of significant DEGs (*Figure 5*). We found that *Npas4* shRNA-upregulated DEGs in the mPFC are also significantly enriched in the DEG module (MB) of the upregulated genes in the PFC of resilience animals (*Bagot et al., 2016*). Although NPAS4 did not influence CSDS-induced social avoidance, these resilience genes could reveal an underlying mechanism to ameliorate or reverse the deficits in reward-related behaviors. Additionally, recent research has shown that analysis of PFC DEGs revealed sex-specific transcriptomic profiles in human depression (*Labonté et al., 2017*), with only 5–10% of genes overlapping between males and females across all brain regions analyzed. This is possibly due to the sex-specific changes in MDD (*Labonté et al., 2017*) as the significant enrichment of ribosomal DEGs was observed only in males, but not females. Although we did not examine sex differences in this study, it will be important to elucidate the NPAS4-mediated transcriptome in females, especially in the mPFC following chronic stress. Of the upregulated DEGs, GO pathway analysis revealed enrichment of genes linked to glutamatergic synaptic transmission and excitability, and PsychENCODE analysis identified a neuronal module of genes linked to glutamatergic excitability that are downregulated in autism spectrum disorders (*Gandal et al., 2018*), suggesting the possibility that CSDS-induced mPFC dendritic spine density loss and excitatory synaptic transmission are produced, in part, by one or more of these synapse-linked genes that are downregulated following stress-induced mPFC NPAS4 expression. Interestingly, 22% of these upregulated DEGs overlapped with NPAS4 target genes identified by ChIP-seq analysis from cultured pyramidal neurons (*Kim et al., 2010*), suggesting that some of the upregulated, synapse-related DEGs could be direct NPAS4 gene targets. Furthermore, we found significant enrichment of DEGs with NPAS4 target genes from an additional NPAS4 ChIP-seq studies (*Brigidi et al., 2019*; *Kim et al., 2010*), suggesting that there may be MDD-related genes directly regulated by NPAS4. Although we did not perform RNA-seq with *Npas4* shRNA after CSDS specifically, this would be an interesting avenue of investigation to determine genes regulated by chronic stress independent of NPAS4. Individual qPCR analyses confirmed the effect of NPAS4 on several genes; however, we saw no effect of acute social defeat stress on the expression of those genes 1 hr after stress exposure (*Figure 5—figure supplement 1*), indicating NPAS4 controls those genes independent of stress exposure. It would be important research to investigate the transcription factor NPAS4-mediated transcriptome (e.g., early and late response genes) in response to stress exposure. In contrast to the upregulated genes, *Npas4* shRNA-downregulated genes showed strong enrichment for ribosomal function and a PsychENCODE module (M15) of excitatory neuron genes associated with ribosome function that is upregulated in ASD and BD, and more

than half of these downregulated *Npas4* shRNA genes are associated with NPAS4 protein (*Figure 5F*). While the functional relevance of ribosome gene enrichment is unclear, the marked enrichment of ribosome-related DEGs is very striking – microarray analysis of blood samples from stress-vulnerable vs. stress-resilient adult human patients found DEGs that were most markedly enriched in ribosome-related pathways and were upregulated based on stress vulnerability (*Hori et al., 2018*). Additionally, RNA-seq analyses from orbitofrontal cortex of postmortem human brains with SCZ, BD, and MDD also identified DEGs enriched for the ribosomal pathway, most of which were upregulated in patient samples (*Darby et al., 2016*). Finally, given that NPAS4 regulates the expression of SST (*Figure 5— figure supplement 1*), it is possible that there are cell extrinsic mechanisms of NPAS4 expression in CaMKIIα neurons on other PFC cell types (i.e., GABAergic interneurons). Therefore, it would be interesting to perform single-cell transcriptomic analysis of PFC tissue following *Npas4* knockdown in excitatory pyramidal neurons following acute or chronic stress, as NPAS4 may influence interneuron (e.g., SST, PV, VIP) gene expression in a non-cell autonomous manner. A remaining question is if lack of NPAS4 in PFC excitatory neurons allows for compensatory increases in other IEGs, such as *Fos* and *Arc*, which could also be answered with a single-cell RNA-sequencing approach. Moreover, it would also be interesting to determine whether NPAS4 overexpression in mPFC enhances CSDS-induced anhedonia-like behavior or a reduction in the proportion of resilience mice.

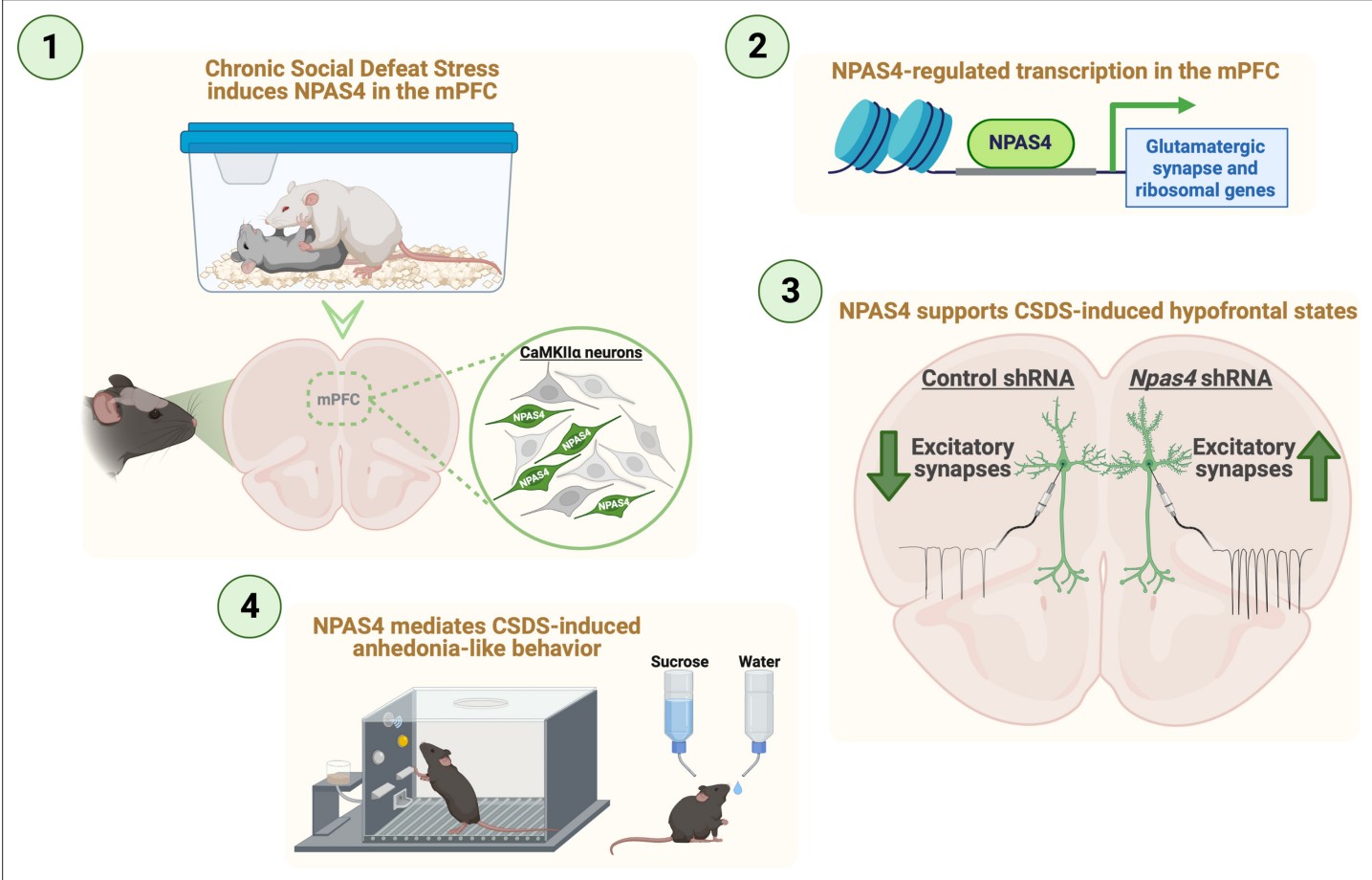

**Figure 6.** Summary for NPAS4 in the medial prefrontal cortex (mPFC) mediates chronic social defeat stress (CSDS)-induced anhedonia-like behavior and reductions in excitatory synapses.

Overall, our findings reveal a novel role for mPFC NPAS4 in CSDS-induced reductions in mPFC pyramidal neuron excitatory synaptic transmission and dendritic spine loss, including the emergence of anhedonia-like behaviors, though mPFC NPAS4 did not impact CSDS-induced social avoidance or anxiety-like behavior (*Figure 6*). We found that mPFC NPAS4 regulates hundreds of genes, including clusters of genes linked to glutamatergic synapse function and ribosomal function, both of which are well-positioned to alter neuronal function. Future strategies targeting these *Npas4*-regulated pathways could be a novel approach to develop therapeutic treatments for hypofrontality and anhedonia-related symptoms in patients struggling with depression, bipolar disorder, and other stress-related neuropsychiatric disorders.

# Materials and methods

## Key resources table

| Reagent type (species) or resource | Designation | Source or reference | Identifiers | Additional information |
|---|---|---|---|---|
| Transfected construct (*Mus musculus*) | AAV2-Anti-Npas4 shRNA | *Taniguchi et al., 2017*, obtained from UNC vector Core and USC vector Core | | |
| Transfected construct (*M. musculus*) | AAV2-Scramble shRNA | *Taniguchi et al., 2017*, obtained from UNC vector Core and USC vector Core | | |
| Biological sample (*M. musculus*) | C57Bl6J mice | The Jackson laboratory | Strain# 000664; RRID:IMSR_JAX:000664 | |
| Antibody | Anti-CaMKIIalpha (mouse monoclonal) | Enzo Life Sciences | Cat# KAM-CA002D; RRID:AB_1659580 | IF(1:1000) |
| Antibody | Anti-somatostatin (rat monoclonal) | Millipore | Cat# MAB354; RRID:AB_2255365 | IF(1:1000) |
| Antibody | Anti- parvalbumin (mouse monoclonal) | Aves | Cat# MAB1572; RRID: AB_2174013 | IF(1:1000) |
| Antibody | Anti-GFP (chicken polyclonal) | Aves | Cat# GFP-1020; RRID: AB_10000240 | IF(1:1000) |
| Antibody | Anti-Npas4 (rabbit polyclonal) | *Lin et al., 2008* | | IF(1:1000–2000) |
| Sequence-based reagent | Scramble shRNA | *Lin et al., 2008* | | GGTTCAGCGTCATAATT TATTCAAGAGATAAATTA TGACGCTGAACC |
| Sequence-based reagent | *Npas4* shRNA | *Lin et al., 2008* | | GGTTGACCCTGATAATT TATTCAAGAGATAAATTA TCAGGGTCAACC |
| Sequence-based reagent | *Npas4* forward primer | *Furukawa-Hibi et al., 2012* | PCR primers | AGCATTCCAGGCT CATCTGAA |
| Sequence-based reagent | *Npas4* reverse primer | *Furukawa-Hibi et al., 2012* | PCR primers | GGCGAAGTAAGT CTTGGTAGGATT |
| Sequence-based reagent | *Npas4* forward primer | *Lin et al., 2008* | PCR primers | GCTATA CTCAGAAGG TCCAGAAGGC |
| Sequence-based reagent | *Npas4* reverse primer | *Lin et al., 2008* | PCR primers | TCAGAGAATGAG GGTAGCACAGC |
| Sequence-based reagent | *Gapdh* forward primer | *Krishnan et al., 2007* | PCR primers | AGGTCGGTGTG AACGGATTTG |
| Sequence-based reagent | *Gapdh* reverse primer | *Krishnan et al., 2007* | PCR primers | TGTAGACCATGT AGTTGAGGTCA |
| Sequence-based reagent | *β-tubulin* forward primer | *Lin et al., 2008* | PCR primers | CGAC AATGAAG CCCTCTACGAC |
| Sequence-based reagent | *β-tubulin* reverse primer | *Lin et al., 2008* | PCR primers | ATGGTGGCAGAC ACAAGGTGGTTG |
| Sequence-based reagent | *cFos* forward primer | *Watanabe et al., 2009* | PCR primers | GTCGACCTAGGG AGGACCTTAC |

*Continued on next page*

*Continued*

| Reagent type (species) or resource | Designation | Source or reference | Identifiers | Additional information |
|---|---|---|---|---|
| Sequence-based reagent | *cFos* reverse primer | *Watanabe et al., 2009* | PCR primers | CATCTCTGGAAG AGGTGAGGAC |
| Sequence-based reagent | *Nfix* forward primer | MGH, Harvard Medical School, Primer Bank | PCR primers | AGCCCCAGCTA CTACAACATA |
| Sequence-based reagent | *Nfix* reverse primer | MGH, Harvard Medical School, Primer Bank | PCR primers | AGTCCAGCTTT CCTGACTTCT |
| Sequence-based reagent | *Sst* forward primer | MGH, Harvard Medical School, Primer Bank | PCR primers | ACCGGGAAAC AGGAACTGG |
| Sequence-based reagent | *Sst* reverse primer | MGH, Harvard Medical School, Primer Bank | PCR primers | TTGCTGGGTT CGAGTTGGC |
| Sequence-based reagent | *Dhcr7* forward primer | ORIGENE | PCR primers, Cat# MP200098 | CAAGACACCAC CTGTGACAGCT |
| Sequence-based reagent | *Dhcr7* reverse primer | ORIGENE | PCR primers, Cat# MP200098 | CTGCTGGAGTAA TGGCACCTTC |
| Sequence-based reagent | *Arpp21* forward primer | ORIGENE | PCR primers, Cat# MP221281 | GGAGTCAGCAAA TACCACAGACC |
| Sequence-based reagent | *Arpp21* reverse primer | ORIGENE | PCR primers, Cat#: MP221281 | CTCCTTGCTGA CTGCTCATCAC |
| Sequence-based reagent | *Hps4* forward primer | ORIGENE | PCR primers, Cat# MP206052 | AGTGTGAACGGA CTGGTGCTGT |
| Sequence-based reagent | *Hps4* reverse primer | ORIGENE | PCR primers, Cat# MP206052 | GTCTCCTTCAGG TGGACTTCCA |
| Sequence-based reagent | *Ache* forward primer | ORIGENE | PCR primers, Cat# MP200188 | TTCCTTCGTGCC TGTGGTAGAC |
| Sequence-based reagent | *Ache* reverse primer | ORIGENE | PCR primers, Cat# MP200188 | CCGTAAACCAGAA AGTAGGAGCC |
| Software, algorithm | HOMER | *Heinz et al., 2010* | | |
| Software, algorithm | STAR | *Dobin et al., 2013* | | |
| Software, algorithm | HTseq | *Anders et al., 2015* | | |
| Software, algorithm | biomaRt | *Durinck et al., 2009* | | |
| Software, algorithm | GOstats | *Falcon and Gentleman, 2007* | | |
| Other | Single-nuclei RNA-seq with mPFC from control C57BL/6J mice | This paper | GSE165586 | snRNA-seq analysis data associated with *Figure 1A-D*. |
| Other | RNA-seq with mPFC from AAV-*Npas4* mRNA shRNA mice | This paper | GSE165586 | RNA-seq analysis data associated with *Figure 5*. |
| Other | ChIP-Seq, NPAS4 | *Brigidi et al., 2019* | GSE127793 | ChIP-seq analysis data associated with *Figure 5E*. |
| Other | ChIP-Seq, NPAS4 | *Kim et al., 2010* | GSE21161 | ChIP-seq analysis data associated with *Figure 5E*. |

## Recombinant plasmids and shRNA expression viral vectors

For knockdown of endogenous *Npas4* mRNA expression in mPFC, a previously validated *Npas4* shRNA, specific to the *Npas4 gene*, or scramble (SC) shRNA control was cloned into the pAAV-shRNA vector as previously described (*Lin et al., 2008*; *Ploski et al., 2011*; *Ramamoorthi et al., 2011*; *Taniguchi et al., 2017*). The adeno-associated virus serotype 2 (AAV2) vector consists of a CMV promoter driving eGFP with a SV40 polyadenylation signal, followed downstream by a U6 RNA polymerase III promoter and *Npas4* shRNA or scrambled (SC) shRNA oligonucleotides, then a polymerase III

termination signal – all flanked by AAV2 inverted terminal repeats. AAV2-*Npas4* shRNA and SC shRNA were processed for packaging and purification by the UNC Vector Core (Chapel Hill, NC).

## Animals

C57BL/6 adult male mice were purchased from Jackson Laboratory (ME) and tested between 8 and 20 weeks of age. Mice were allowed access to food and water ad libitum and were kept on a 12 hr light-dark cycle. All procedures were in accordance with Institutional Animal Care and Use (IACUC) guidelines.

## Viral-mediated gene transfer

Stereotaxic surgery was performed under general anesthesia with a ketamine/xylazine cocktail (120 mg/kg: 16 mg/kg) or isoflurane (induction 4% v/v, maintenance 1–2% v/v). Coordinates to target the mPFC (ventral portion of cingulate, prelimbic, and infralimbic cortices) were +1.85–1.95 mm anterior, +0.75 mm lateral, and 2.65–2.25 mm ventral from bregma (relative to skull) at a 15° angle in all mice (*Covington et al., 2010*). AAV2-scramble (SC) shRNA (2.9 * 10^9 and 1.1 * 10^12 GC/mL) and AAV2-*Npas4* shRNA (4.3 * 10^9 and 3.1 * 10^12 GC/mL) were delivered using Hamilton syringes or nanoinjectors with pulled glass capillaries at a rate of 0.1 µL/min for 0.4 µL total at the dorsoventral sites, followed by raising the needle and an additional 0.4 µL delivery of virus. After waiting for an additional 5–10 min, needles were completely retracted. Viral placements were confirmed through immunohistochemistry for bicistronic expression of eGFP from the AAV2 viral vectors by experimenters blinded to the experimental conditions. Animals with off-target virus infection or no infection in one or both hemispheres were excluded from the analysis of behavioral phenotypes.

## Single-nuclei RNA-seq and bioinformatic analysis

Control C57BL/6J mice (Vgat-cre positive) were live-decapitated at 8 weeks of age. Brains were rapidly extracted into a supplemented 4°C Hibernate A medium, with GlutaMAX supplement (Fisher), B27 supplement (Fisher), and NxGen RNase inhibitor (0.2 U/uL, Lucigen), and incubated for 30 s. The brain was sectioned into 1 mm sections using a brain matrix. The prefrontal cortex (PFC) was dissected with fine forceps, flash frozen on dry ice, and stored at –80°C. The nuclear isolation protocol was modified from *Savell et al., 2020*. On the day of nuclear dissociation, the frozen tissue was slowly thawed on ice and chopped with a scalpel blade 100 times in two orthogonal directions on a glass Petri dish. The chopped tissue of three PFCs were pooled and transferred to a hypotonic lysis buffer (10 mM Tris-HCl, 10 mM NaCl, 3 mM MgCl2, 0.1% IGEPAL [Sigma] in Nuclease-Free Water). Supplemented Hibernate A medium was added, and the tissue was triturated ~30 times/sample using three glass pipettes of decreasing diameter. The tissue was then filtered using a 40 micron filter (Fisher). The nuclei were isolated using 500 × $g$ centrifugation and washed with 1× PBS + 1.0% BSA and 0.2 U/uL RNase inhibitor. Nuclei were incubated with 7-aminoactinomycin D (7-AAD) (Invitrogen) for 5 min and sorted for 7-AAD-positive single nuclei using fluorescence-activated nuclear sorting on the Aria II. Samples were counted and diluted to 1500 nuclei/uL before immediate processing on the 10X Genomics Single-Cell Protocol by the MUSC Translation Science Lab. Libraries were constructed using the Chromium Single Cell 3' Library Construction Kit (10X Genomics, v3.1) and sequenced at Vanderbilt's Next Gen Sequencing Core (Illumina NovaSeq 6000). Raw sequencing data were processed with Cell Ranger (v6.1.2) (PMID: 28091601). Cellranger mkfastq command was used to demultiplex the different samples and cellranger count command was used to generate gene–cell expression matrices. Ambient RNA contamination was inferred and removed using CellBender (v0.232) with standard parameters. Mouse genome mm10 was used for the alignment, and genecode vM25 was used for gene annotation and coordinates (*Frankish et al., 2021*).

## Chronic social defeat stress

CSDS was performed as previously described (*Golden et al., 2011*; *Krishnan et al., 2007*). CD1 retired male breeders (Charles River Laboratory, CA) were single-housed for 3–5 days before CSDS procedures to establish their territorial cage, then pre-screened for aggressive behavior. Experimental C57BL/6J male mice were introduced to the aggressor's territorial cage, physically contacted and attacked by the aggressor for 5–10 min, and then separated by a clear plastic board with multiple small holes for 24 hr. Experimental mice were introduced to a new CD1 aggressor each day. The no

stress control mice were housed with another non-stressed C57BL/6J male mouse, separated by the same plastic board, and the cage partner was changed every day for 10 days of the CSDS experiment.

## qRT-PCR for gene expression

Brain tissue was collected at the described time point after social defeat stress, or no stress control condition, and kept frozen at –80°C until processed for the following steps. RNA isolation, reverse transcription, and quantitative real-time PCR were carried out as described previously (*Taniguchi et al., 2017*). Mouse tissue samples were homogenized in QIAzol solution and processed for RNA purification using the miRNeasy kit (QIAGEN, MD) following the manufacture's protocol. The total RNA was reverse transcribed using SuperScript III (Invitrogen) with a random hexamer primer following the manufacturer's instructions. Quantitative PCR (qPCR) was performed using SYBR Green (Bio-Rad, CA). The level of mRNA expression was analyzed by the fold change relative to *Gapdh* or *β-tubulin* expression. The relative mRNA level was analyzed as the difference from experimental condition relative to controls. Please see Key Resources Table for primer sequences.

## Immunohistochemistry

Mouse brains were fixed overnight in 4% PFA in 1× PBS and transferred to a 30% sucrose solution in 1× PBS before slicing (40 or 50 μm) with a microtome. The slices were permeabilized and blocked in 3% BSA, 0.3% Triton X-100, 0.2% Tween-20, 3% normal donkey or goat serum in PBS, then incubated with primary antibodies: anti-GFP (1:1000, Aves Labs, Inc, OR; 1:1000–10,000, Invitrogen), anti-NPAS4 (1:1000, rabbit, kindly provided by Dr. Michael Greenberg's lab), anti-CaMKIIα (1:1000, Enzo, NY, 6G9), anti-somatostatin (SST) (1:1000, Millipore MAB354), and anti-parvalbumin (PV) (1:1000, Millipore MAB1572) in blocking buffer at room temperature for 2–4 hr or at 4°C overnight. Following a series of 1× PBS rinses, slices were incubated for 1–3 hr at room temperature with secondary antibodies (donkey anti-rabbit 488, goat anti-mouse 594, donkey anti-mouse Cy3, or donkey anti-chicken 488) while protected from light. Slices were counterstained with Hoechst, mounted, and coverslipped on glass slides using AquaMount (Thermo Scientific, MA) or ProlongGold (Thermo Scientific, MA) and analyzed with confocal microscopy (Zeiss LSM 880). The expression level of NPAS4 protein in each cell was measured using ImageJ software in CaMKIIα-positive cells under experimenter-blinded conditions.

## Social interaction assay

Social interaction (SI) assay was performed as previously described (*Golden et al., 2011*; *Krishnan et al., 2007*). The social interaction assay was performed 24 hr after the last CSDS procedure. The assay was performed in an open-field arena (44 cm × 44 cm) with the social target's holding cage. The time mice spent in the interaction zone (8 cm from the social target) was examined for 5 min in the absence and then the presence of a novel CD1 mouse, of which the experimental animal never met, under dim red light using AnyMaze 5.1 (Stoelting Co, Wood Dale) or Ethovision 3.0 software (Noldus, Leesburg, VA). The social interaction ratio was calculated as the time spent in the social interaction zone in the presence of an interaction partner divided by the time in the absence.

## Sucrose preference test

The sucrose preference procedure was performed as previously described (*Taniguchi et al., 2012*). Single-housed mice were provided with Division of Laboratory Animal Resources-approved tap water in two identical double-ball-bearing sipper-style bottles for 2 days, followed by 2 days of 1% (w/v) sucrose solution in tap water to allow for acclimation. Mice were then given one bottle containing tap water and another containing the 1% sucrose solution. Consumption from each bottle was measured every 24 hr for 4 days, and bottle positions were swapped each day to avoid potential side bias. The sucrose preference was calculated by percentage of 1% sucrose consumption volume divided by total liquid consumption volume (sucrose + tap water). Measurement of liquid consumption volume was performed with experimenters blinded to conditions.

## Elevated plus maze

The elevated plus maze (EPM) was conducted under the bright light (80 Lux at the closed arm) as previously performed (*Taniguchi et al., 2017*). Mice were positioned in the center of the maze, and

behavior was recorded by video tracking using AnyMaze 5.1 or Ethovision 3.0 software as previously performed (*Penrod et al., 2019*). The time spent in the open arms was recorded for 5 min.

## Sucrose self-administration assay

Sucrose self-administration was conducted for 2 hr at the same time each day for 9–10 days of acquisition training, followed by a progressive ratio schedule as described previously (*Taniguchi et al., 2017*). Briefly, sucrose availability was signaled both by the house light and a light above the active nosepoke hole. Following a poke in the active hole, both availability lights went off and a cue light inside the nose poke hole was illuminated. Sucrose pellets were delivered immediately upon the active nosepoke, followed by a 10 s time-out period. Nose pokes in the inactive hole were without programmed consequences. During a progressive ratio schedule of reinforcement, the requirements for a sucrose delivery were increased on a subsequent sucrose delivery in an exponential manner. Animals were allowed to self-administer until they failed to earn a sucrose pellet in a 60 min time frame. The last sucrose pellet achieved is reported as an indicator of how much animals consumed before reaching breakpoint.

## Dendritic spine morphometric analyses

Mouse brains were collected with rapid live decapitation 24 hr after the social interaction assay and fixed overnight in 4% PFA in 1× PB, then transferred to a 30% sucrose solution in 1× PB before slicing (40 µm) with a vibratome. Deep layer eGFP-expressing pyramidal neurons in the prelimbic cortex were sampled for dendritic spine analyses as described previously (*Siemsen et al., 2019*). Briefly, proximal apical dendrites were imaged with a Leica SP8 laser scanning confocal microscope equipped with HyD detectors for enhanced sensitivity. Dendritic spine segments were selected only if they satisfied the following criteria: (1) could clearly be traced back to a cell body of origin, (2) were not obfuscated by other dendrites, and (3) were proximal to the branch point separating the apical tuft from the proximal apical dendrite. Images were collected with a ×63 oil immersion objective (1.4 N.A.) at 1024 × 1024 frame size, 4.1× digital zoom, and a 0.1 µm Z-step size (0.04 × 0.04 × 0.1 µm voxel size). Pinhole was set at 0.8 airy units and held constant. Laser power and gain were empirically determined and then held relatively constant, only adjusting to avoid saturated voxels. Huygens Software (Scientific Volume Imaging, Hilversum, NL) was used to deconvolve 3D Z-stacks. Deconvolved Z-stacks were then imported into Imaris (version 9.0.1) software (Bitplane, Zurich, CH). The filament tool was then used to trace and assign the dendrite shaft. Dendritic spines were then semi-automatically traced using the autopath function, and an automatic threshold was used to determine dendritic spine head diameter. Variables exported included the average spine head diameter (in µm) as well as the number of dendritic spines per µm of dendrite (spine density). 3–10 segments were sampled per animal, and the average spine head diameter and the spine density were calculated for each segment. Data for each variable was then expressed as number of spine segments/number of animals. All analyses were performed under experimenter-blinded conditions.

## Electrophysiology

All acute-slice electrophysiological experiments were performed in SC and *Npas4* shRNA[PFC] mice at 12–14 weeks old. Acute coronal slices (300 µm thickness) containing mPFC were prepared in a semi-frozen 300 mOsM dissection solution containing (in mM): 100.0 choline chloride, 2.5 KCl, 1.25 $Na_2H_2PO_4$, 25.0 $NaHCO_3$, 25.0 D-glucose, 3.1 Na-pyruvate, 9.0 Na-ascorbate, 7.0 $MgCl_2$, 0.5 $CaCl_2$ and 5.0 kynurenic acid (pH 7.4) and was continually equilibrated with 95% $O_2$ and 5% $CO_2$ prior to and during the slicing procedure. Slices were transferred to a 315 mOsM normal artificial cerebrospinal fluid (ACSF) solution containing (in mM): 127.0 NaCl, 2.5 KCl, 1.2 $Na_2H_2PO_4$, 24.0 $NaHCO_3$, 11.0 D-glucose, 1.2 $MgCl_2$, 2.4 $CaCl_2$, and 0.4 Na-ascorbate (pH 7.4) to recover at 37°C for 30 min, and then transferred to room temperature ACSF for an additional 30 min prior to recording.

mPFC pyramidal neurons of layer 5 were visualized with infrared differential interference contrast optics (DIC/infrared optics) and identified by their location, apical dendrites, and spiking patterns in response to depolarizing current injection and AAV2-mediated SC shRNA or *Npas4* shRNA expression cells were identified by expression of GFP. Unless stated otherwise, all electrophysiological experiments were performed in whole-cell voltage-clamp mode at –70 mV using borosilicate pipettes (4–6 MΩ) made on NARISHIGE puller (NARISHIGE, PG10) from borosilicate tubing (Sutter Instruments)

and filled by an internal solution containing (in mM): 140.0 CsMetSO$_4$, 5.0 KCl, 1.0 MgCl$_2$, 0.2 EGTA, 11 HEPES, 2 NaATP, 0.2 Na$_2$GTP (pH 7.2; 290–295 mOsm).

The AMPA-receptor-mediated mEPSCs were recorded in the presence of 100 µM picrotoxin (GABAARs antagonist, Sigma-Aldrich) and TTX (sodium channels blocker, Sigma-Aldrich). Data were recorded in a series of 10 traces (sweeps), 10 s each. At the beginning of each sweep, a depolarizing step (4 mV for 100ms) was generated to monitor series (10–40 MΩ) and input resistance (>400 MΩ). To analyze data, synaptic events were detected via custom parameters in MiniAnalysis software (Synaptosoft, Decatur, GA) and subsequently confirmed by the observer blinded to the experimental conditions. Data were measured until 700 events in a series were analyzed, or until the maximal duration of the series.

Paired EPSC for PPR measurements were generated at –70 mV with the inter-stimulus interval of 50 ms at frequency of 0.05 Hz – 3 stimulus in 1 min. The peak amplitude of the second EPSC (P2) was divided by the peak of the first amplitude (P1) to generate the PPR (P2/P1).

All data (recordings) were acquired and analyzed by amplifier AXOPATCH 200B (Axon Instruments), digitizer BNC2090 (National instruments), and software AxoGraph v1.7.0, Clampfit v8.0 (pClamp, Molecular Devices), and MiniAnalysis Program v6.0.9 (Synaptosoft). Data were filtered at 2 kHz via AXOPATCH 200B amplifier (Axon Instruments) and digitized at 20 kHz via AxoGraph v1.7.0.

## RNA-seq and bioinformatic analysis

Total RNA was isolated from AAV2-mediated eGFP-positive mPFC slices using the QIAGEN RNA purification kit, as described above. Sequencing was performed by BGI genomics using PolyA mRNA isolation, directional RNA-seq library preparation, and a BIGSeq-500 sequencer. Reads were aligned to the mouse mm10 reference genome using STAR (v2.7.1a) (*Dobin et al., 2013*). Only uniquely mapped reads were retained for further analyses. Quality control metrics were assessed by Picard tool (RRID:SCR_006525) (http://broadinstitute.github.io/picard/). Gencode annotation for mm10 (version M21) was used as reference alignment annotation and downstream quantification. Gene-level expression was calculated using HTseq (v0.9.1) (*Anders et al., 2015*) using the intersection-strict mode by exon. Counts were calculated based on protein-coding genes from the annotation file.

## Differential gene expression

Counts were normalized using counts per million reads (CPM). Genes with no reads were removed. Differential expression analysis was performed in R using linear modeling as following: lm(gene expression ~ Treatment + Batch). We estimated log2 fold changes and p-values. p-Values were adjusted for multiple comparisons using a Benjamini–Hochberg correction (FDR). Differentially expressed genes were analyzed at FDR <0.05. Mouse Gene IDs were translated into Human Gene IDs using the biomaRt package (v2.46.0) in R (*Durinck et al., 2009*).

## Gene ontology analyses

The functional annotation of differentially expressed and co-expressed genes was performed using GOstats (*Falcon and Gentleman, 2007*). A Benjamini–Hochberg FDR (FDR <0.05) was applied as a multiple comparison adjustment.

## Gene set enrichment

Gene set enrichment was performed in R using Fisher's exact test with the following parameters: alternative = 'greater,' confidence level = 0.95. We reported odds ratio (OR) and Benjamini–Hochberg adjusted p-values (FDR).

## Statistics

One-way, two-way, and three-way analyses of variance (ANOVAs) with or without repeated-measures (RM) were used, followed by Bonferroni or Tukey post hoc tests when a significant interaction was revealed, to analyze mRNA expression, number of NPAS4 (+) cells, NPAS4 protein expression in each cell, percentage of CaMKIIα(+) cells, social interaction, social aversion, sucrose preference, elevated plus maze, sucrose self-administration acquisition and discrimination, dendritic spine morphometric data, and breakpoint in the progressive ratio test. All statistics were performed using GraphPad Prism, except SPSS software was used to handle complex datasets (e.g., three-way ANOVAs). Statistical

outliers were detected using a Grubbs test and excluded from analysis. All data are presented as the mean ± SEM. Significance was shown as *p<0.05, **p<0.01, ***p<0.001, ****p<0.0001, and nonsignificant values were either not noted or shown as n.s.

## Acknowledgements

The authors thank Yingxi Lin, Rachel Penrod, Laura Smith, Yuhong Guo, Ben Zirlin, and Sara Pilling in the Taniguchi and Cowan lab for technical assistance, comments on the manuscript, and helpful discussions. BWH was supported by an NIH predoctoral fellowship (F31 DA048557 and T32 DA07288). BMS was supported by an NIH postdoctoral fellowship (F32 DA050427). MT was supported by a NARSAD Young Investigator Award from the Brain & Behavior Research Foundation (Grant #22765). This work was supported by grants from the NIH (UL1 TR001450 to MT, R01 DA032708 to CWC, and DA046373 to CWC and MT).

## Additional information

### Funding

| Funder | Grant reference number | Author |
|---|---|---|
| National Institutes of Health | F31 DA048557 | Brandon W Hughes |
| National Institutes of Health | T32 DA07288 | Brandon W Hughes |
| National Institutes of Health | F32 DA050427 | Benjamin M Siemsen |
| Brain and Behavior Research Foundation | F32 DA050427 | Makoto Taniguchi |
| National Institutes of Health | UL1 TR001450 | Makoto Taniguchi |
| National Institutes of Health | R01 DA032708 | Christopher W Cowan |
| National Institutes of Health | DA046373 | Christopher W Cowan Makoto Taniguchi |

The funders had no role in study design, data collection and interpretation, or the decision to submit the work for publication.

### Author contributions

Brandon W Hughes, Conceptualization, Data curation, Formal analysis, Validation, Investigation, Methodology, Writing - original draft, Writing - review and editing; Benjamin M Siemsen, Michael D Scofield, Data curation, Investigation, Methodology; Evgeny Tsvetkov, Data curation, Formal analysis, conducted the electrophysiology recording experiments and analyzed the data for Figure 4C-4H in the revision; Stefano Berto, Jordan S Carter, Data curation; Jaswinder Kumar, Data curation, Methodology; Rebecca G Cornbrooks, Rose Marie Akiki, Data curation, Investigation; Jennifer Y Cho, Data curation, conducted the single nuclei RNA-seq sample preparations for Figure 1A-1D in the revision; Kirsten K Snyder, Data curation, conducted an immunohistochemistry experiment and analyzed the data for Figure 1H in the revision; Ahlem Assali, Data curation, conducted the single nuclei RNA-seq sample preparations for Figure 1A-1D in the revision; Christopher W Cowan, Conceptualization, Formal analysis, Supervision, Funding acquisition, Writing - original draft, Project administration, Writing - review and editing; Makoto Taniguchi, Conceptualization, Resources, Data curation, Formal analysis, Supervision, Funding acquisition, Validation, Investigation, Visualization, Methodology, Writing - original draft, Project administration, Writing - review and editing

### Author ORCIDs

Brandon W Hughes http://orcid.org/0000-0002-2742-6453
Benjamin M Siemsen http://orcid.org/0000-0002-6105-6054

Christopher W Cowan ⬭ http://orcid.org/0000-0001-5472-3296
Makoto Taniguchi ⬭ http://orcid.org/0000-0001-6356-2463

### Ethics

All experimental procedures were in accordance with Institutional Animal Care and Use (IACUC) guidelines and approved protocols #01156 of the Medical University of South Carolina.

### Decision letter and Author response

Decision letter https://doi.org/10.7554/eLife.75631.sa1
Author response https://doi.org/10.7554/eLife.75631.sa2

## Additional files

### Supplementary files

• Supplementary file 1. RNA-seq data Related to *Figure 5*.

• Transparent reporting form

• Source data 1. Detailed statistics information related to *Figures 1–5* and figure supplements.

### Data availability

Sequencing data have been deposited in GEO under accession codes GSE165586. The data access is limited until the finding is accepted for publication.

The following dataset was generated:

| Author(s) | Year | Dataset title | Dataset URL | Database and Identifier |
| --- | --- | --- | --- | --- |
| Taniguchi M, Hughes B, Cowan C | 2021 | RNA sequencing approaches to identify the differential expression gene in the mPFC with AAV-mediated Npas4 shRNA expression | https://www.ncbi.nlm.nih.gov/geo/query/acc.cgi?acc=GSE165586 | NCBI Gene Expression Omnibus, GSE165586 |

The following previously published datasets were used:

| Author(s) | Year | Dataset title | Dataset URL | Database and Identifier |
| --- | --- | --- | --- | --- |
| Labonte B, Engmann O, Purushothaman I, Ménard C, Wang J, Tan C, Scarpa JR, Moy G, Loh YE, Cahill M, Lorsch ZS, Hamilton PJ, Calipari ES, Hodes GE, Issler O, Kronman H, Pfau M, Obradovic A, Dong Y, Neve RL, Russo S, Kazarskis A, Tamminga C, Mechawar N, Turecki G, Zhang B, Shen L, Nestler EJ | 2017 | Sex-specific Transcriptional Signatures in Human Depression | https://www.ncbi.nlm.nih.gov/geo/query/acc.cgi?acc=GSE102556 | NCBI Gene Expression Omnibus, GSE102556 |
| Brigidi GS, Hayes MGB, Delos Santos NP, Hartzell AL, Texari L, Lin PA, Bartlett A, Ecker JR, Benner C, Heinz S, Bloodgood BL | 2019 | Characterization of Npas4 and heterodimer DNA binding in stimulated and silenced rat neurons | https://www.ncbi.nlm.nih.gov/geo/query/acc.cgi?acc=GSE127793 | NCBI Gene Expression Omnibus, GSE127793 |

*Continued on next page*

*Continued*

| Author(s) | Year | Dataset title | Dataset URL | Database and Identifier |
|---|---|---|---|---|
| Kim TK, Hemberg M, Gray JM, Costa AM, Bear DM, Wu J, Harmin DA, Laptewicz M, Barbara-Haley K, Kuersten S, Markenscoff-Papadimitriou E, Kuhl D, Bito H, Worley PF, Kreiman G, Greenberg ME | 2010 | Widespread transcription at neuronal activity-regulated enhancers | https://www.ncbi.nlm.nih.gov/geo/query/acc.cgi?acc=GSE21161 | NCBI Gene Expression Omnibus, GSE21161 |

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
