## [Editor Report]

This important manuscript shows compelling evidence for a role for the transcription factor NPAS4 in the medial prefrontal cortex in regulating stress-induced behavior, pyramidal neuron spine density, and gene expression. There is beautiful depth of mechanistic insight into how chronic stress produces anhedonia-like behavior. This paper will be of interest to the field of stress neurobiology and neuropsychiatry.

---

## [Decision Letter]

**Decision letter after peer review:**

Thank you for submitting your article "NPAS4 in the medial prefrontal cortex mediates chronic social defeat stress-induced anhedonia and dendritic spine loss" for consideration by *eLife*. Your article has been reviewed by 3 peer reviewers, and the evaluation has been overseen by Kate Wassum as the Senior and Reviewing Editor. The following individuals involved in review of your submission have agreed to reveal their identity: Scott J Russo (Reviewer #2); Catherine Jensen Peña (Reviewer #3).

This is a very interesting manuscript that shows a role for mPFC NPAS4 in regulating stress induced behavior, pyramidal neuron spine density, and gene expression. We appreciated that the manuscript is well written, the rigorous statistical approach, and novel results. We have a concerns about some of the conclusions and the depth of some of the analyses that should be addressed in a revision. We think addressing these concerns is likely to require additional experiments. Below are the essential revisions.

Essential revisions:

1. The abstract and results both state that reward motivation is altered in the NPAS4 shRNA group. However, figure 3 does not convincingly show this. There is no interpretable effect of shRNA on sucrose delivery, discrimination, or breakpoint. The authors claim that the significant correlation by linear regression between breakpoint and SI ratio in control but not NPAS4 shRNA groups can be interpreted as a change in reward motivation, and they use an appropriate ANCOVA analysis to show that these two factors are correlated in the control but not the shNPAS4 group. However, the interpretation of this lack of correlation is unclear. If the other data show that NPAS4 knockdown has no effect on SI or operant sucrose consumption, how can an effect on the correlation between these behaviors be meaningful? If the authors are stating that the lack of NPAS4 prevents the mice with a social withdrawal phenotype from developing an anhedonic breakpoint phenotype, wouldn't it be equally valid to say that lack of NPAS4 causes mice with no social withdrawal phenotype to develop an anhedonic breakpoint phenotype? And what would either conclusion mean for our understanding of the mechanisms of stress effects on these behaviors? Moreover, how do we relate these correlations or lack of correlations with supplemental figure 2, which shows no correlation between SI ratio and sucrose preference in any group?

Additional data is needed to either more convincingly support the conclusion that CSDS produced anhedonia and deficits in motivation that are rescued by Npas4 manipulation or to provide a clearer behavioral anchor for the interaction between CSDS and mPFC Npas4.

2. Why not stratify "susceptible" and "resilient" mice after CSDS? Does Npas4 correlate with susceptibility? Authors found that among CSDS-exposed mice, those "susceptible" according to social interaction also showed lower reward motivation, whereas resilient mice had higher reward motivation, and Npas4 knockdown in PFC reversed/broke this correlation. Perhaps a stratification of susceptible and resilient (which may require more N) would help provide a cleaner behavioral effect?

3. Admittedly it's difficult to target subregions of the mPFC in mice (i.e., infralimbic v. prelimbic) but we wonder if viral placement can explain the high variability observed in behavior? For example, when analyzing viral expression did the authors notice whether individual differences in viral expression patterns (ie. more intense expression in one subregion over another) correlate with differences behavior? This may be particularly relevant for data from the effort based task in Figure 3.

4. The authors use shRNA to knockdown NPAS4 expression in mPFC and find interesting effects on sucrose preference. However, it is possible that these effects are mediated by the shRNA targeting genes other than NPAS4 or that the role of mPFC NPAS4 expression in this behavior is indirect, and NPAS4 itself does not drive the anhedonic response to stress. Validation of the efficacy and selectively of the Npas4 shRNA approach in mPFC needs to be demonstrated or reference to this in the literature more clearly described.

5. The results of the RNAseq experiments are interesting, but lack depth. Typically, some validation of at least a few of the putative target genes is required to confidently interpret the results of an unbiased sequencing experiment. qPCR for individual target genes is a good start, but ChIP experiments to show that at least some of the putative NPAS4 targets are actually bound by NPAS4 in mPFC would be stronger, and showing by ChIP that this binding is regulated by CSDS would be ideal.

As presented, it seems as though sequencing was only in control mice, though given the protective effect of Npas4 knockdown it would be interesting to see the normal and lack of transcriptional effect after CSDS with knockdown – especially given NPAS4's role in experience/activity-dependent transcriptional regulation. This would significantly enhance the manuscript. But if it is not feasible, then considerable discussion should be given to this limitation.

6. A change in total dendritic spine number is an interesting initial finding, but this portion of the study lacks depth. The fact that there is no change in spine head diameter suggests that there may be a change in synapse number without change in synapse strength. Therefore, functional analyses of glutamatergic synapses (mEPSPs, AMPA/NMDA ratio, etc) examining the effects of NPAS4 knockdown or overexpression would improve the study substantially. If this is not possible, then some discussion of this possibility and limitation is warranted.

7. RNA seq and qPCR data from Figure 5 and supplement Figure 4 suggests that NPAS4 regulates somatostatin (Sst). Given that Sst neurons don't express NPAS4, this would suggest a cell extrinsic mechanism (ie. changes in NPAS4 in CAMK^+^ cells alters SST neurons). Can the authors comment on this in the discussion? Are there other genes from interneurons that seem to be regulated by such extrinsic mechanisms. Is there a way to filter the data sets by cell type (ie. pyramidal neurons versus interneurons) so as to infer cell type specific expression across all transcripts?

8. Given recent improvements in CSDS across sexes, more of a rationale is needed for the exclusions of females from this study. This major limitation of the manuscript also needs to be discussed.

9. If not interneurons, what other cell types in mPFC account for the 20-25% of cells that express NPAS4 but not CAMK^+^? Is NPAS4 upregulated by stress in them? Also, it would be better to show the data in Figure 1D as NPAS4 expression within each cell type instead of % of cellular marker. This way the reader can see that NPAS4 is only upregulated in CAMK^+^ cells.

10. One has to wonder when we suppress IEGs and TF's in general what's really happening. Does this just dampen transcriptional response broadly? Does activation of other IEGs make up for lack of NPAS4? Is there something special about NPAS4 or would knocking down Fos have the same effect? Note that additional experiments to answer these broad questions are not expected.

11. In addition to the very helpful tables, please report all statistics in the main manuscript, consistent with *eLife*'s policy: https://reviewer.elifesciences.org/author-guide/full "Report exact p-values wherever possible alongside the summary statistics and 95% confidence intervals. These should be reported for all key questions and not only when the p-value is less than 0.05."

[Editors' note: further revisions were suggested prior to acceptance, as described below.]

Thank you for resubmitting your work entitled "NPAS4 in the medial prefrontal cortex mediates chronic social defeat stress-induced anhedonia-like behavior and reductions in excitatory synapses" for further consideration by *eLife*. Your revised article has been evaluated by Kate Wassum (Senior and Reviewing Editor).

The manuscript has been improved but there are some remaining issues that need to be addressed, as outlined below:

Reviewer 3 has two points that should be straightforward to address:

– Please consider discussing the fact that the qPCR validation only validated Npas4 itself, and there was no effect of stress on the other genes.

– The figures also indicate a main effect of Npas4 KD for other genes, but statistics are missing from this section of the results.

– Please include summary statistics (e.g., t, F values) and degrees of freedom in your statistical reporting in the main manuscript (results or figure legends).

*Reviewer #1 (Recommendations for the authors):*

The authors did an outstanding job of addressing the most pressing concerns of both reviews. The inclusion of new behavioral and electrophysiological data fills in critical gaps in the original manuscript, and the main interpretations are now well-supported by the data. I feel the manuscript will make an important contribution to the field, and I recommend it be published in its current form.

*Reviewer #2 (Recommendations for the authors):*

I believe the authors have adequately revised the manuscript. I have no further concerns.

*Reviewer #3 (Recommendations for the authors):*

The authors have done a substantial amount of work on this manuscript, including new experiments and analyses. The new patch-clamp experiments in particular add substantial depth and mechanism to the study. The relationship between mPFC Npas4 and behavior is more convincing.

What do the authors make of the fact that the qPCR validation only validated Npas4 itself, and there was no effect of stress on the other genes? The figures also indicate a main effect of Npas4 KD for other genes, but statistics are missing from this section of the results.

Related to the response to reviewer comment 2: Equal numbers of susceptible and resilient mice are not needed to stratify susceptible vs resilient or perform a correlation analysis; nevertheless, I don't feel this is required.

Overall, I think the revised figures, discussion, and response to reviewers improve the depth of the science and better contextualize the findings and approaches.

---

## [Author Response]

Essential revisions:1. The abstract and results both state that reward motivation is altered in the NPAS4 shRNA group. However, figure 3 does not convincingly show this. There is no interpretable effect of shRNA on sucrose delivery, discrimination, or breakpoint. The authors claim that the significant correlation by linear regression between breakpoint and SI ratio in control but not NPAS4 shRNA groups can be interpreted as a change in reward motivation, and they use an appropriate ANCOVA analysis to show that these two factors are correlated in the control but not the shNPAS4 group. However, the interpretation of this lack of correlation is unclear. If the other data show that NPAS4 knockdown has no effect on SI or operant sucrose consumption, how can an effect on the correlation between these behaviors be meaningful? If the authors are stating that the lack of NPAS4 prevents the mice with a social withdrawal phenotype from developing an anhedonic breakpoint phenotype, wouldn't it be equally valid to say that lack of NPAS4 causes mice with no social withdrawal phenotype to develop an anhedonic breakpoint phenotype? And what would either conclusion mean for our understanding of the mechanisms of stress effects on these behaviors? Moreover, how do we relate these correlations or lack of correlations with supplemental figure 2, which shows no correlation between SI ratio and sucrose preference in any group?Additional data is needed to either more convincingly support the conclusion that CSDS produced anhedonia and deficits in motivation that are rescued by Npas4 manipulation or to provide a clearer behavioral anchor for the interaction between CSDS and mPFC Npas4.2. Why not stratify "susceptible" and "resilient" mice after CSDS? Does Npas4 correlate with susceptibility? Authors found that among CSDS-exposed mice, those "susceptible" according to social interaction also showed lower reward motivation, whereas resilient mice had higher reward motivation, and Npas4 knockdown in PFC reversed/broke this correlation. Perhaps a stratification of susceptible and resilient (which may require more N) would help provide a cleaner behavioral effect?

We thank the reviewer for these helpful comments and suggestions. In the revised manuscript, we have now added three new cohorts of animals for CSDS and the subsequent behavioral assays (Figure 2C-H; Figure 3A-D). With more statistical power, we observed that *Npas4* shRNA in the mPFC produced a main effect of *Npas4* in the sucrose PR breakpoint analysis (Figure 3C). Post-hoc analysis, given the a priori hypothesis that stress-susceptible animals show distinct deficits in behavior (Krishnan et al., 2007, *Cell*), identified that CSDS-susceptible animals expressing *Npas4* shRNA^PFC^ exhibited a significantly higher sucrose PR breakpoint when compared to controls (Figure 3D), suggesting that NPAS4 in PFC of stress-susceptible animals functions to suppress motivation to seek natural rewards. In these new additional cohorts, we had a number of animals that were severely injured by CSDS, and our veterinary staff requested that we discontinue with those mice, so we were unable to continue through the behavioral battery. In other words, although some animals exhibited the stress-susceptible phenotype in the social interaction assay, we were unable to continue with the sucrose SA assay due to ethical concerns. Other factors likely contributed to the low proportion of susceptible mice, including the biological sex of the investigator. Several cohorts of CSDS were conducted by the female investigator, that demonstrated a significantly lower proportion of susceptible mice compared to the experiment performed by male investigators (Georgiou et al., *Nature Neuroscience*, 2022). These scenario contributed to the lower sample numbers in the susceptible animals in the sucrose SA tests, even after adding multiple new cohorts. Unfortunately, we did not obtain an equal number of susceptible and resilient animals, despite our valiant efforts to increase the power in the susceptible category to enable stratification and statistical analysis of SI subgroups.

3. Admittedly it's difficult to target subregions of the mPFC in mice (i.e., infralimbic v. prelimbic) but we wonder if viral placement can explain the high variability observed in behavior? For example, when analyzing viral expression did the authors notice whether individual differences in viral expression patterns (ie. more intense expression in one subregion over another) correlate with differences behavior? This may be particularly relevant for data from the effort based task in Figure 3.

Our AAV2 and stereotaxic targeting coordinates produced strong viral-mediated gene expression throughout both the infralimbic (IL) and prelimbic cortex (PrL). For those injections that produced a slight subregion bias, we did not detect any obvious differential effects on behavior. As the differential functions of PrL and IL subregions are well documented in reward-related behaviors, it would be an interesting future direction to pursue, and we have included additional text in the revised discussion to address this possibility and limitation.

4. The authors use shRNA to knockdown NPAS4 expression in mPFC and find interesting effects on sucrose preference. However, it is possible that these effects are mediated by the shRNA targeting genes other than NPAS4 or that the role of mPFC NPAS4 expression in this behavior is indirect, and NPAS4 itself does not drive the anhedonic response to stress. Validation of the efficacy and selectively of the Npas4 shRNA approach in mPFC needs to be demonstrated or reference to this in the literature more clearly described.

We agree that off-target effects of an shRNA are always of concern. The *Npas4* shRNA utilized in our study has been used frequently to reduce NPAS4 expression, and it has been validated in multiple experimental contexts (Lin et al., 2008; Ramamoorthi et al., 2011; Ploski et al., 2011; Taniguchi et al., 2017). Previous studies, including ours, have observed similar effects of *Npas4* shRNA and conditional knockout (*Npas4*^fl/fl^+Cre) mice in electrophysical recordings of hippocampal slices and in cocaine-conditioned behaviors in the NAc, and while we could repeat all of our experiment with a second shRNA or floxed Npas4 mice, we feel the expense and animal life costs are too great. However, we now reference these prior studies, and we denote in the revised discussion that we can’t rule out off-target effects contributing to the observed phenotypes.

5. The results of the RNAseq experiments are interesting, but lack depth. Typically, some validation of at least a few of the putative target genes is required to confidently interpret the results of an unbiased sequencing experiment. qPCR for individual target genes is a good start, but ChIP experiments to show that at least some of the putative NPAS4 targets are actually bound by NPAS4 in mPFC would be stronger, and showing by ChIP that this binding is regulated by CSDS would be ideal.As presented, it seems as though sequencing was only in control mice, though given the protective effect of Npas4 knockdown it would be interesting to see the normal and lack of transcriptional effect after CSDS with knockdown – especially given NPAS4's role in experience/activity-dependent transcriptional regulation. This would significantly enhance the manuscript. But if it is not feasible, then considerable discussion should be given to this limitation.

To partially address the reviewer’s concern, we now compare our RNA-seq data with previously published NPAS4 ChIP-seq data (Figure 5E) (Kim et al., 2010; Brigidi et al., 2019). The new analysis reveals significant enrichment of NPAS4 genomic binding at many of our DEGs, which suggests that many of the observed DEGs are likely to be *direct* NPAS4 target genes (Figure 5).We agree that a new ChIP-seq study combined with a new RNA-seq study (+/- acute vs. chronic social defeat stress) with mPFC tissues is the ideal future direction for this study. Part of the challenge with this approach is the uncertainty about the optimal time point after stress (acute or chronic) since NPAS4 is rapidly and transiently produced by stress so analysis at multiple time points (e.g., 5 mins, 15 mins, 1 hr, and 24 hrs) +/- acute vs. chronic social defeat will be required to truly understand direct vs. indirect targets genes. We now comment on this limitation of our current data sets in the revised discussion, and we hope the reviewer will agree that addressing this issue is beyond the scope of our data-rich first report on mPFC NPAS4 in CSDS-induced anhedonia-like behavior.

6. A change in total dendritic spine number is an interesting initial finding, but this portion of the study lacks depth. The fact that there is no change in spine head diameter suggests that there may be a change in synapse number without change in synapse strength. Therefore, functional analyses of glutamatergic synapses (mEPSPs, AMPA/NMDA ratio, etc) examining the effects of NPAS4 knockdown or overexpression would improve the study substantially. If this is not possible, then some discussion of this possibility and limitation is warranted.

We agree that functional synaptic analysis is important. In the revised manuscript we now add new patch-clamp recording data from mPFC deep-layer pyramidal neurons. We observe a significant role for NPAS4 on glutamatergic synapse function. Consistent with the stress-induced decrease in dendritic spine density, we find that CSDS produced a significant increase in the inter- event interval of mEPSCs in the SC shRNA control mice, and this mini frequency effect was blocked by *Npas4* shRNA^PFC^. In addition, *Npas4* shRNA significantly increased mEPSC amplitude independent of CSDS. Using evoked EPSCs, we found that CSDS significantly increased the paired-plus ratio in the shRNA controls, but *Npas4* shRNA blocked this effect (Figure 4G-H), revealing an NPAS4-dependent effect of CSDS to reduce presynaptic function. Since NPAS4 shRNA is expressed in the postsynaptic cell being recorded, it seems likely that this CSDS and NPAS4 effect is cell non-autonomous; however, some glutamatergic inputs to the mPFC pyramidal neuron might come from local inputs that also express NPAS4 shRNA. Understanding the underlying stress and NPAS4-dependent mechanisms will be an important future direction. Together, our new functional excitatory synapse analyses (Figure 4C-F) extend the structural synapse loss documented in the original manuscript and provide additional insights into NPAS4’s role in both “no stress” and CSDS conditions (Figure 4B).

7. RNA seq and qPCR data from Figure 5 and supplement Figure 4 suggests that NPAS4 regulates somatostatin (Sst). Given that Sst neurons don't express NPAS4, this would suggest a cell extrinsic mechanism (ie. changes in NPAS4 in CAMK^+^ cells alters SST neurons). Can the authors comment on this in the discussion? Are there other genes from interneurons that seem to be regulated by such extrinsic mechanisms. Is there a way to filter the data sets by cell type (ie. pyramidal neurons versus interneurons) so as to infer cell type specific expression across all transcripts?

We agree that functional synaptic analysis is important. In the revised manuscript we now add new patch-clamp recording data from mPFC deep-layer pyramidal neurons. We observe a significant role for NPAS4 on glutamatergic synapse function. Consistent with the stress-induced decrease in dendritic spine density, we find that CSDS produced a significant increase in the inter- event interval of mEPSCs in the SC shRNA control mice, and this mini frequency effect was blocked by *Npas4* shRNA^PFC^. In addition, *Npas4* shRNA significantly increased mEPSC amplitude independent of CSDS. Using evoked EPSCs, we found that CSDS significantly increased the paired-plus ratio in the shRNA controls, but *Npas4* shRNA blocked this effect (Figure 4G-H), revealing an NPAS4-dependent effect of CSDS to reduce presynaptic function. Since NPAS4 shRNA is expressed in the postsynaptic cell being recorded, it seems likely that this CSDS and NPAS4 effect is cell non-autonomous; however, some glutamatergic inputs to the mPFC pyramidal neuron might come from local inputs that also express NPAS4 shRNA. Understanding the underlying stress and NPAS4-dependent mechanisms will be an important future direction. Together, our new functional excitatory synapse analyses (Figure 4C-F) extend the structural synapse loss documented in the original manuscript and provide additional insights into NPAS4’s role in both “no stress” and CSDS conditions (Figure 4B).

8. Given recent improvements in CSDS across sexes, more of a rationale is needed for the exclusions of females from this study. This major limitation of the manuscript also needs to be discussed.

We completely agree, and we would love to repeat all of our key findings in a chronic stress design that allows for inclusion of females in a future study. We now clearly acknowledge in the revised manuscript (paragraph 2 of the discussion) that the male-only findings are a major limitation of our current study. In the 11 years since we began this NPAS4/CSDS project, there have been several competing approaches that allow female analysis, but we’ve had very limited success getting them to work reliably and convincingly, which is a common refrain in conversations with other CSDS investigators we’ve encountered. We hope to shift to a different chronic stress model in the future to extend our current findings and compare possible sex-effects.

9. If not interneurons, what other cell types in mPFC account for the 20-25% of cells that express NPAS4 but not CAMK^+^? Is NPAS4 upregulated by stress in them? Also, it would be better to show the data in Figure 1D as NPAS4 expression within each cell type instead of % of cellular marker. This way the reader can see that NPAS4 is only upregulated in CAMK^+^ cells.

We appreciate the great suggestion and have revised Figure 1D indulging adding Figure 1H. We now also include recent snRNA-seq analysis of *Npas4* mRNA expression in mPFC tissues of C57BL/6J mice (Figure 1A-D). Similar to BrainRNAseq.org, we observe that Npas4-positive neurons in mPFC snRNA-seq are predominantly excitatory neuron clusters (~92%) with the remainder found in multiple interneuron types, including PV, Sst, and Adarb2 clusters (Figure 1C-1D). As such, we strongly suspect that the ~20% NPAS4+ cells that did not co-localize with CaMKIIα IHC staining represent neurons with subthreshold levels of CaMKIIα (below antibody detection) and/or are interneuron populations not measured (e.g., Adarb2). Future scRNA-seq studies examining acute and chronic social defeat stress at various time points after the stress experiences and with cell type-specific NPAS4 manipulations will be important to fully understand NPAS4’s cell type- specific distribution and roles in stress-induced anhedonia.

10. One has to wonder when we suppress IEGs and TF's in general what's really happening. Does this just dampen transcriptional response broadly? Does activation of other IEGs make up for lack of NPAS4? Is there something special about NPAS4 or would knocking down Fos have the same effect? Note that additional experiments to answer these broad questions are not expected.

The question raised by the reviewer is very interesting. Our data showed that the reduction of NPAS4 increased the mEPSC amplitude and upregulated genes involved in glutamatergic synapses (e.g. *Arc)*, suggesting that reduction of NPAS4 enhanced, directly or indirectly, IEG transcriptional responses. While we did not compare +/- CSDS, under basal conditions it appears that suppression of NPAS4 promotes IEG expression (e.g., *Fos* and *Arc*). We note that this observation is in contrast to a report in hippocampal cultured neurons where loss of NPAS4 reduced broad IEG induction following KCl depolarization (Ramamoorthi et al., 2011). This could represent differences in developing vs. adult neurons, hippocampus vs. cortex, or culture vs. in vivo study designs.

11. In addition to the very helpful tables, please report all statistics in the main manuscript, consistent with eLife's policy: https://reviewer.elifesciences.org/author-guide/full "Report exact p-values wherever possible alongside the summary statistics and 95% confidence intervals. These should be reported for all key questions and not only when the p-value is less than 0.05."

We now provide all statistical information in the revised manuscript.

[Editors' note: further revisions were suggested prior to acceptance, as described below.]

The manuscript has been improved but there are some remaining issues that need to be addressed, as outlined below:Reviewer 3 has two points that should be straightforward to address:– Please consider discussing the fact that the qPCR validation only validated Npas4 itself, and there was no effect of stress on the other genes.

We have revised the Discussion section to reflect the results that NPAS4 impacts several genes, but there was no effect from acute social defeat stress on the expression of those genes. In the revised manuscript, we have discussed the need for future transcriptome analysis to investigate NPAS4-mediated transcriptome in response to stress.

– The figures also indicate a main effect of Npas4 KD for other genes, but statistics are missing from this section of the results.

We now provide statistical information of main effect of *Npas4* KD in the revised manuscript.

– Please include summary statistics (e.g., t, F values) and degrees of freedom in your statistical reporting in the main manuscript (results or figure legends).

We provide summary statistics (t and F values, and degrees of freedom) information in the revised main manuscript.